# "Gene accordions" cause genotypic and phenotypic heterogeneity in clonal populations of *Staphylococcus aureus*

Darya Belikova[1], Angelika Jochim[1], Jeffrey Power[1], Matthew T. G. Holden [2] & Simon Heilbronner [1,3,4 ✉]

Gene tandem amplifications are thought to drive bacterial evolution, but they are transient in the absence of selection, making their investigation challenging. Here, we analyze genomic sequences of *Staphylococcus aureus* USA300 isolates from the same geographical area to identify variations in gene copy number, which we confirm by long-read sequencing. We find several hotspots of variation, including the *csa1* cluster encoding lipoproteins known to be immunogenic. We also show that the *csa1* locus expands and contracts during bacterial growth in vitro and during systemic infection of mice, and recombination creates rapid heterogeneity in initially clonal cultures. Furthermore, *csa1* copy number variants differ in their immunostimulatory capacity, revealing a mechanism by which gene copy number variation can modulate the host immune response.

[1] Interfaculty Institute of Microbiology and Infection Medicine, Department of Infection Biology, University of Tübingen, Tübingen, Germany. [2] School of Medicine, University of St Andrews, St Andrews KY16 9TF, UK. [3] German Centre for Infection Research (DZIF), Partner Site Tübingen, Tübingen, Germany. [4] (DFG) Cluster of Excellence EXC 2124 Controlling Microbes to Fight Infections, Tübingen, Germany. ✉email: simon.heilbronner@uni-tuebingen.de

Within their natural environment, prokaryotes are constantly exposed to changing conditions ranging from shifting temperatures and changing nutrient availabilities to fluctuating levels of noxious compounds. The tremendous ability of prokaryotes to adapt to environmental changes is due to their capacity to alter their genetic material rapidly, which is a key element of their evolutionary success.

Prokaryotic genomes show a high degree of plasticity and acquisition of genetic traits by horizontal gene transfer (HGT) is well studied[1]. However, HGT relies on appropriate genetic material to be externally available and will be hampered if the bacterial community under selection is rather homogenous. Alternatively, genomic diversity created by single nucleotide polymorphisms (SNPs) can facilitate adaptation processes[2]. The same is true for genomic rearrangements that impact expression levels of genes[3,4]. Rearrangements occur in most cases stochastically by recombination between homologous DNA motifs and allow inversions as well as deletions or tandem amplifications of genetic material[3]. Tandem arrays of genes are most frequently caused by a RecA-dependent mechanism known as gene duplication and amplification (GDA). For the development of GDAs, the "accordion" model is well accepted and proposes that initial duplications can arise in RecA-dependent or independent fashions[5–7]. After the primary duplication event, long perfect tandem repeats allow RecA-dependent amplification or, conversely, the loss of the duplication (segregation) at high rate[6,7]. Due to the high frequency of repetitive DNA segments in prokaryotic chromosomes gene copy number variants caused by GDAs create genetic and phenotypic heterogeneity in prokaryotic populations[8–11]. Selective pressures can favor certain copy number variants allowing stabilization of the arrays within the population. This phenomenon is often observed in the context of antibiotic resistance[6,12]. For instance, Nicoloff et al. have recently demonstrated that GDAs cause antibiotic resistant subpopulations in otherwise sensitive populations in many clinically relevant species[13]. However, expansion and contraction of gene arrays should also harbor the potential to shape populations under unclear constrains such as pathogens or commensals facing a multitude of host-associated selective pressures. Therefore, the analysis of tandem amplifications in pathogens might pinpoint genetic loci under evolutionary pressure in the host. However, apart from loci encoding malfunctioning resistance determinants, it remains unclear whether special genomic regions are particularly prone to tandem amplification in the presence of environmental triggers such as antibiotic pressure or host immune defences.

The invasive pathogen Staphylococcus aureus is a major cause of healthcare and community-associated infections leading to severe morbidity and mortality. S. aureus shows a remarkable ability to adapt to the healthcare setting where strong artificial selective pressures such as antibiotics and disinfectants drive the evolution of pathogens to develop resistance[14]. In the age of next generation sequencing (NGS) thousands of genomes of strains from many different pathogenic species, including S. aureus, have been sequenced. This opens plentiful opportunities to identify gene copy number variations caused by GDAs between closely related strains and to link variation to phenotypic characteristics using experimental approaches.

Here, we test this approach using previously published NGS datasets of clinical populations of S. aureus USA300 from the urban area of New York city[15]. Our analysis reveals frequent gene copy number variations in loci that harbor repetitive sequences. Some of the proteins encoded at these loci have previously been linked to host colonization and virulence such as the surface-anchored molecule SdrD and the Spl serine proteases. Most prominent is copy number variation within the lipoprotein gene array csa1. Using experimental approaches, we find amplification of csa1 and sdrD to occur readily in vitro. The frequency of amplification is increased 10-fold when RecA is induced by the fluoroquinolone antibiotic ciprofloxacin, supporting the "accordion" model of amplification. csa1 copy number variants show distinct differences in Csa1 protein levels and altered immunostimulatory activity suggesting roles for the proteins in the interaction with the immune system. Using systemic models of invasive disease, we find that csa1 copy number variation also occurrs in vivo with a higher frequency than observed in any in vitro experiment. This depends on functional intact csa1 coding sequences with associated protein expression, suggesting that environmental constrains favor the creation of genotypic and phenotypic heterogeneity amongst clonal populations in vivo.

## Results

**Gene copy number variation is frequently observed in staphylococcal chromosomes.** We thought to investigate whether gene copy number variation caused by GDAs in repetitive parts of the genomes creates unrecognized heterogeneity in S. aureus populations. In order to identify GDAs we focused on a published set of S. aureus USA300 genome sequences[15] from New York that were obtained using Illumina HiSeq-technology which allows smooth coverage and accurate scaffolding. The short read datasets from 348 strains were mapped to the USA300 reference sequence FPR3757[16]. Coverage across the chromosome was analyzed using a minimum window size of 100 bp and areas showing ≥2× coverage were regarded as putatively amplified regions. We also included areas showing no coverage, which represent deletions. We focused on the core genome and the pathogenicity islands vSaα and vSaβ but excluded genes associated with other mobile genetic elements (MGEs) identified for USA300 (phages φSA2usa, φSA3usa, SCCmec, and transposases[16]) as differences in coverage of these will in part reflect similar MGEs inserted into various sites in the chromosome.

We found several areas of the core genome that varied in depth of coverage (Supplementary Data 1). These loci harbored highly repetitive DNA motifs, supporting the hypothesis that RecA-dependent recombination might have created copy number variation. We discriminated three different types of repetitive elements facilitating recombination (Fig. 1a). Firstly, repetitive motifs were present as domains within several genes of a tandem array. The sdrCDE locus encodes three cell wall-anchored proteins with highly repetitive serine-aspartate (SD) repeats (85.9–88.3% identity between the genes). We identified 24 isolates lacking either sdrD or sdrE or both. All these deletions could be explained by recombination between the SD-encoding regions (Supplementary Fig. 1a, Supplementary Data 1). Secondly, repetitive domains were present within a single protein coding sequence (CDS). The surface-anchored protein SasG harbors highly repetitive G5-E domains[17]. The G5-E-encoding DNA was frequently overrepresented/deleted in individual isolates, suggesting that recombination altered the size of the open reading frame (Supplementary Fig. 1b, Supplementary Data 1). Finally, we realized that several S. aureus loci encode tandem arrays of genes that are highly similar over the entire length of the CDS. Amongst those was the array of serine proteases (splABCDEF) and the superantigen-like toxins (ssl) (Supplementary Data 1). However, most prominently associated with copy number variation in our set of isolates was the locus encoding surface displayed proteins previously named "conserved staphylococcal antigens 1" (csa1)[18]. The csa1 genes encode lipoproteins belonging to a group known as tandem-lipoproteins (Lpps). Four loci encoding similar Lpps are present in the S. aureus chromosome (csa1, the lipoprotein-like genes (lpl) encoded on the pathogenicity island vSaα[19], and two further loci here referred to as lpp3 and lpp4, respectively) (Supplementary

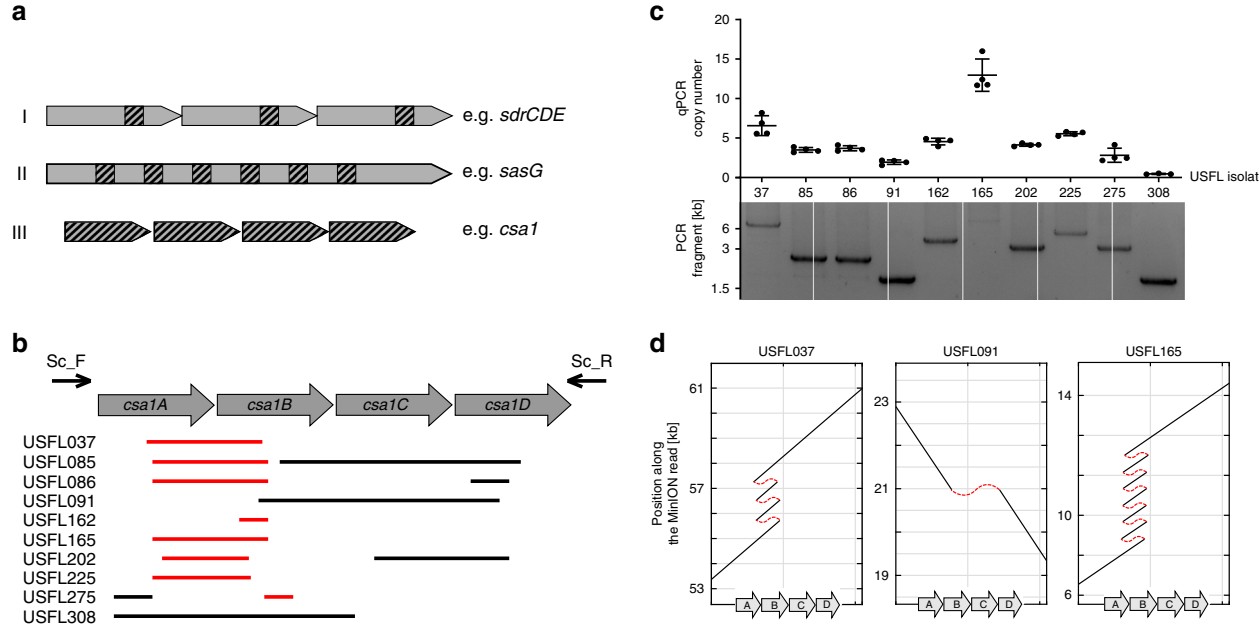

**Fig. 1 csa1 gene copy number variation within the csa1 locus of S. aureus USA300 isolates. a** Schematic diagram of the three categories of repetitive sequences creating copy number variation. Hatched areas denote regions of homology. **b** Schematic representation of the deletions and putative amplifications of csa1 in different isolates. Coding sequences are indicated. Red and gray lines represent increased coverage (>2 fold) or deletions within the USFL isolates, respectively. **c** The upper panel shows the csa1 copy number of the indicated clinical isolates as measured by qPCR. Mean and SD of four replicate qPCRs on a single DNA isolation is shown. The lower panel shows DNA fragments amplified by conventional PCR using primers csa1_Sc.F and csa1_Sc.R indicated in (**b**). The experiment was performed once using the same batch of DNA used for qPCR. Source data are provided as a Source Data file. **d** USFL isolates were sequenced using MinION technology. For each sequenced isolate, a single long read that is aligned against the csa1 locus of USA300 FPR3757 is shown. Tandem amplifications manifest as multiple regions within the read with homology to csa1, allowing calculation of gene copy number. Hatched red lines emphasize that a single long read mapping repeatedly to the csa1 genes is analyzed. Genes csa1A, csa1B, csa1C, and csa1D are indicated. Shown are read_ch:51530_110 (USFL037); read_ch:10493_161 (USFL091); read_ch:40122_41 (USFL165). Source data are provided as a Source Data file.

Fig. 2). All *lpp* genes exhibit 46.1–81.9% identity. In the USA300 FPR3757 genome the *csa1*, *lpl*, *lpp3*, and *lpp4* loci harbor four, ten, one, and three genes, respectively (Supplementary Fig. 2). Of note, despite the strong homology among all genes, two of them (*lpp4*C (SAUSA300_2424) and *lpp3* (SAUSA300_0205)) do not encode lipoboxes, suggesting that the proteins are not anchored to the membrane[20]. Occasional deletions were observed in all loci in individual isolates but only *csa1* showed putative amplifications in 80% of the clinical isolates (Fig. 1, Supplementary Data 1). The four *csa1* genes of FPR3757 (*csa1A-D*) comprise 771, 771, 771, and 768 bp, respectively, and exhibit 61–80% identity (Supplementary Fig. 2). Each of the genes was occasionally deleted in individual isolates. In contrast, amplifications covered exclusively the *csa1A-csa1B* genes (Fig. 1b).

We assumed that the observed differences in NGS scaffolding indicated tandem amplification events within the array rather than additional copies of the genes on cryptic plasmids or on additional sites of the chromosome. To validate this, the *csa1ABCD*, *sdrCDE*, and *sasG* loci of various isolates were amplified by PCR (Fig. 1c, lower panel, and Supplementary Fig. 1). Size differences between the amplified fragments were in agreement with predicted deletions and tandem amplifications. Of note, strain USFL165 harbored a *csa1* locus that could not be amplified by PCR. We performed MinION long read sequencing of selected strains and extracted individual reads covering the *csa1* array (Fig. 1d). This analysis confirmed tandem arrays of 7, 3, and 10 copies of *csa1* genes in USFL037, USFL091, and USFL165, respectively. Of note, analysis of single reads identifies the gene copy number carried by an individual cell of the culture. However, since tandem arrays of genes are intrinsically unstable,

individual cells might differ in copy number. Therefore, we designed qPCR primers to amplify a conserved fragment of all four *csa1* genes simultaneously. Using chromosomal DNA as template, qPCR analysis allows us to determine the average gene copy number within the population. qPCR results correlated well with long read sequencing and identified 7 ± 1, 2 ± 0.5, and 13 ± 2 copies of *csa1* genes in USFL037, USFL091, and USFL165, respectively (Fig. 1c, upper panel).

**Amplification occurs constantly and the frequency is increased by antibiotic pressure.** We sought to investigate the development of GDAs during growth of a single clone and chose the *csa1ABCD* locus as it showed frequent variation among the clinical isolates. We introduced a tetracycline (Tc) resistance determinant (*tetK*) between *csa1B* and *csa1C* in USA300 LAC by allelic replacement. TetK is known to specify a resistance level that is gene-dosage dependent[21]. The wild type was phenotypically sensitive to 2 μg/ml Tc. The USA300 *csa1::tetK* strain was resistant to 2 μg/ml Tc and displayed weak growth at Tc concentrations up to 10 μg/ml while growth was completely inhibited at concentrations exceeding 10 μg/ml. We anticipated that spontaneous amplifications of *csa1* would also span *tetK* thereby increasing Tc resistance and providing a selectable phenotype. We grew USA300 *csa1::tetK* over three consecutive days in six parallel broth cultures in the absence of Tc. Therefore, amplification within the broth culture was not favored by antibiotic selection, allowing estimation of stochastic amplification in the absence of selection. Each day, the cultures were plated on agar plates containing 20 μg/ml Tc and arising resistant colonies were picked to analyze the *csa1* copy number by qPCR (Figs. 2 and 3). Not all Tc-resistant clones

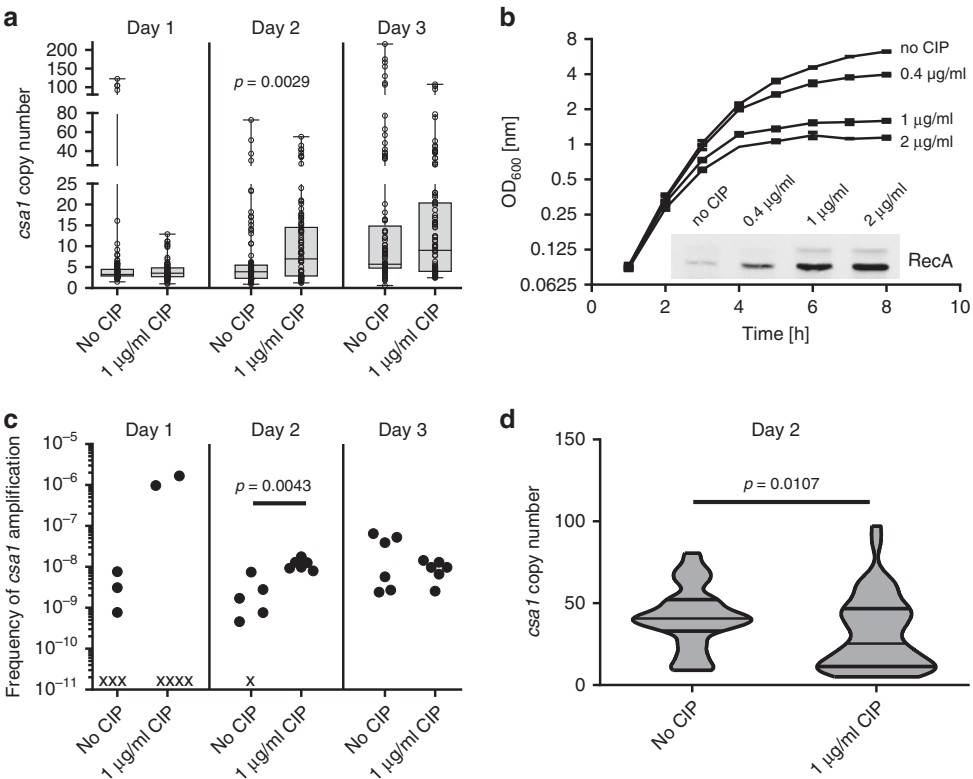

**Fig. 2 Copy number diversification after in vitro evolution. a** USA300 *csa1::tetK* was grown over three consecutive days in six parallel cultures in the presence or absence of ciprofloxacin (CIP). Each day, the copy number of up to 16 clones of each culture showing high Tc resistance was screened by qPCR. Upper and lower box limits and the horizontal lines within the boxes represent 25 and 75% percentiles and the medians, respectively. The whiskers of the plots indicate minimum and maximum range. All data points are shown. Data are derived from six independent experiments and represent: Day 1, $n = 96$ and $n = 73$ of "no Cip" and "1 μg/ml Cip", respectively; Day 2, $n = 93$ and $n = 96$ of "no Cip" and "1 μg/ml Cip", respectively; Day 3, $n = 93$ and $n = 96$ of "no Cip" and "1 μg/ml Cip", respectively. Datasets were not normal distributed (D'Agostino & Pearson omnibus test <0.0001) and statistical analysis was performed using two-tailed Mann–Whitney test. Source data are provided as a Source Data file. **b** Growth curves of USA300 *csa1::tetK* in TSB containing various concentrations of the antibiotic ciprofloxacin (CIP), which is known to enhance RecA expression. As expected, at increasing inhibitory concentrations of CIP, the growth rate of USA300 *csa1::tetK* decreased as assessed by measuring $OD_{600}$ every 2 h. This was accompanied by an increase in RecA protein levels as assessed in mid-exponential cells by Western blot analysis using LI-COR infrared technology. A representative blot is shown wherein the lower and upper bands represent RecA and an unspecific signal, respectively. Mean and SEM of three independent experiments are shown. Source data are provided as a Source Data file. **c** Frequency of amplification (Tc20-resistant clones showing at least a 2 fold increase in *csa1* copy number compared to the parental strain by qPCR) within the total population of living cells. Shown are the results of six independent experiments. *x* indicates a culture in which amplification was not detected. Statistical analysis was performed using two-tailed Mann–Whitney test. Source data are provided as a Source Data file. **d** A USA300 *csa1::tetK* high copy number variant (harboring ~50 copies of *csa1*) was grown over two consecutive days in three parallel cultures in the presence or absence of CIP. Each day, the copy number of the strains of 17 randomly chosen clones was screened by qPCR ($n = 51$). Shown is a violin plot, length of the box indicates minimum and maximum range. Width of the bar indicates accumulation of data points. Horizontal lines within the boxes represent 25 and 75% percentiles and the medians, respectively. Statistical analysis was performed using two-tailed Mann–Whitney test. Source data are provided as a Source Data file.

showed elevated *csa1* copy numbers for reasons that were not apparent. Nevertheless, increased Tc resistance correlated frequently with an increased *csa1* gene copy number and numbers as high as 100–200 copies were detected several times (Fig. 2a). After 24 h of growth, strains with amplifications in the *csa1* locus were isolated in 2 of 6 parallel cultures. This number increased to 5/6 and 6/6 cultures after two and three days of growth, respectively. The copy numbers ranged from 4 to ~200 *csa1* genes with copy numbers as high as 100 detected at day 1 (Fig. 2a). Also, the level of copy number diversity increased over time with $I_{50}$-values (copy-number variation of 50% of the population surrounding the median) of 1.84 at day 1 and 3.24 and 10.36 at days 2 and 3, respectively (Fig. 2a).

To confirm tandem amplification of *csa1* within such strains, we used the MinION technology to sequence four independently evolved isolates that displayed high copy number as measured by

qPCR (C6 - $149 \pm 22$ copies; E28 - $78 \pm 10$; L38 - $64 \pm 3$; III37 - $26 \pm 2$) (Fig. 3a). From the MinION reads we extracted those that covered the *csa1* gene array as shown in Fig. 3b. For strains C6 and E28 this approach produced individual reads (70–120 kb in length) covering exclusively the *csa1* array but lacking upstream or downstream sequences, confirming the presence of tandem amplifications too large to be completely covered (Fig. 3b). This made precise copy number determination impossible but revealed that strains C6 and E28 harbored at least 77 and 72 copies, respectively. Analysis of strains L38 and III37 revealed single reads covering upstream and downstream sequences as well as the *csa1* array, suggesting 56 and 23 copies of the *csa1* gene. However, for both strains also reads lacking upstream and downstream sequences were identified suggesting that the sequenced populations were heterogeneous with respect to copy numbers of tandem amplifications and that some cells within the L38 and

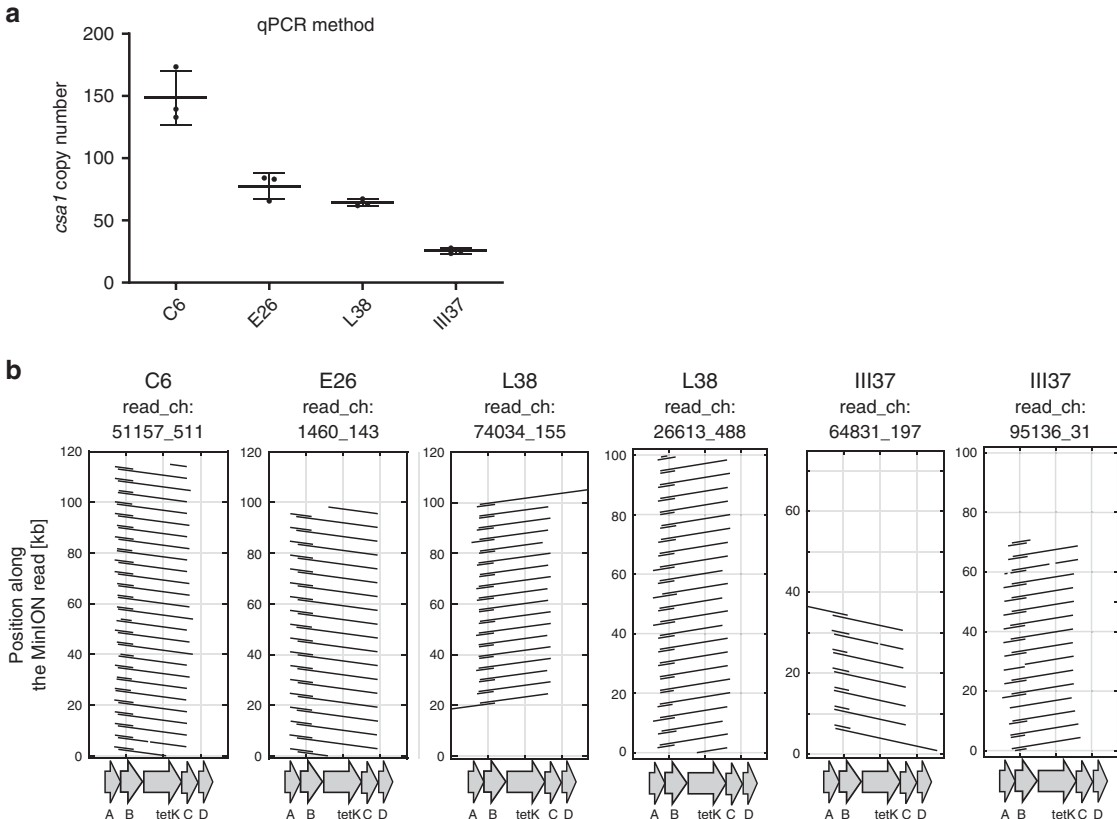

**Fig. 3 Tandem amplification of *csa1::tetK*. a** *csa1* copy number of four independent, high Tc-resistant strains measured by qPCR. Mean and SD of three replicate qPCRs on a single DNA isolation is shown. Source data are provided as a Source Data file. **b** MinION sequence analysis of high Tc-resistant isolates. For each sequenced isolate, single reads covering the *csa1* array were aligned against the *csa1::tetK* locus of the parental strain. Tandem amplifications manifest as multiple regions within the read with homology to *csa1::tetK*. Connecting lines within the read are omitted for reasons of clarity. Genes *csa1A, csa1B, tetK, csa1C,* and *csa1D* are indicated. Shown are read_ch:51157_511 (C6), read_ch:1460_143 (E28), read_ch:74034_155 (L38), and read_ch: 64813_197 (III37). Source data are provided as a Source Data file.

III37 isolates harbored at least 68 and 43 copies of *csa1* genes, respectively (Fig. 3b).

Due to the high sequence similarity of the *csa1* genes, we speculated that extension of the array is mediated by the SOS recombinase RecA. Fluoroquinolone antibiotics such as ciprofloxacin (CIP) are known to induce RecA expression[22]. Subinhibitory concentrations of CIP increased cellular RecA levels in a dose-dependent manner (Fig. 2b). This led to a more rapid diversification of the culture with differences compared to untreated cultures being most prominent at day two. This was reflected both in the level of diversity ($I_{50}$ values of CIP-treated cultures increased from 2.401 at day 1 to 11.931 and 16.68 at day 2 and day 3, respectively) and in the frequency of strains harboring *csa1* amplifications, which was ~10 fold higher in CIP-treated compared to untreated cultures at day two (Fig. 2c). Only on day 3 the diversity of untreated cultures approached that of CIP-treated cultures.

Amplification and segregation events are two sides of the same coin. Amplification of gene arrays extends the length of sequence homology thereby increasing the frequency of recombination which leads to further diversification. To investigate the stability of the tandem arrays, we started an in vitro evolution experiment with a variant containing ~50 copies. After two consecutive passages in liquid broth the cultures were plated on agar without Tc to allow growth of all variants and the *csa1* copy number of randomly chosen colonies was determined. The distribution of copy numbers varied significantly. The untreated culture displayed three distinct populations. ~50% of isolates gathered

tightly around the median *csa1* copy number (40.7 copies) while additional populations harboring 52–81 copies and 9–32 copies were observed (Fig. 2d). In contrast, the distribution of copy numbers within the population of the CIP-treated cultures was more diverse. The median copy number was decreased to 25.3 but a clustering around the median was not observed. In contrast clustering of the population around the 25th percentile, which dropped from 32.96 to 11.34 upon CIP treatment, was observed. These results are in line with the "accordion"-model of GDAs leading to diversification of the population in high and low copy number variants with low copy numbers being favored under nonselective conditions. We speculated that our culture conditions were nonselective regarding the function of the Csa1 proteins and reflected stochastic creation of heterogeneity. To test this, we constructed the isogenic strain *csa1*(FS)*::tetK*. This strain carries a *tetK*-labeled *csa1* locus where each gene is inactivated by a single nonsense mutation (Supplementary Fig. 3a). Passaging of this strain in the presence or absence of CIP revealed a similar pattern of amplification as observed for the functional locus, supporting the idea that Csa1 function was not selected for under our experimental conditions (Supplementary Fig. 3b).

Our bioinformatic analysis revealed frequent copy number variation also within the *sdrCDE* locus in which recombination between SD repeat-encoding regions seemed to create frequent deletions. Again amplifications and segregations go hand in hand but in many cases deletions might be detected more conveniently as they represent the dead end of the GDA-mechanism due to their irreversible nature. Therefore, we hypothesized that *sdrCDE*

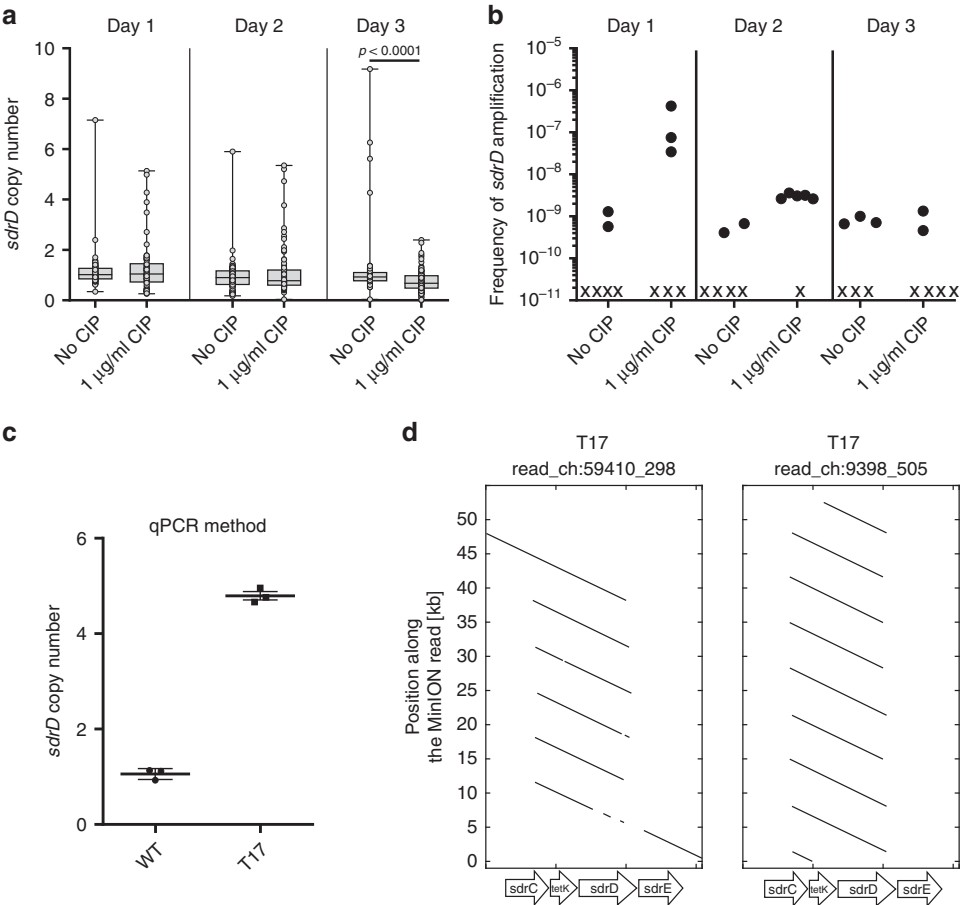

**Fig. 4 Tandem amplification of *sdrD::tetK*. a** USA300 *sdrD::tetK* was grown over three consecutive days in six parallel cultures in the presence or absence of ciprofloxacin (CIP). Each day, the copy number of up to 16 clones of each culture showing high Tc resistance was screened by qPCR. Upper and lower box limits and the horizontal lines within the boxes represent 25 and 75% percentiles and the medians, respectively. The whiskers of the plots indicate minimum and maximum range. Data are derived from six independent experiments and represent: Day 1, $n = 87$ and $n = 76$ of "no Cip" and "1 µg/ml Cip", respectively; Day 2, $n = 90$ and $n = 95$ of "no Cip" and "1 µg/ml Cip", respectively; Day 3, $n = 78$ and $n = 96$ of "no Cip" and "1 µg/ml Cip", respectively. All data points are shown. Datasets were not normal distributed (D'Agostino & Pearson omnibus test <0.0001) and statistical analysis was performed using two-tailed Mann–Whitney test. Source data are provided as a Source Data file. **b** Frequency of amplification (Tc$_{20}$-resistant clones showing at least a 2 fold increase in *csa1* copy number compared to the parental strain by qPCR) within the total population of living cells. Shown are the results of six independent experiments. *x* indicates a culture in which amplification was not detected. Source data are provided as a Source Data file. **c** *sdrD* copy number of a high Tc-resistant strain in comparison to the parental strain measured by qPCR. Mean and SD of three replicate qPCRs on a single DNA isolation is shown. Source data are provided as a Source Data file. **d** MinION sequence analysis of the high Tc-resistant isolate. Single reads covering the *sdrCDE* array were aligned against the *sdrD::tetK* locus of the parental strain. Tandem amplifications manifest as multiple regions within the read with homology to *sdrD::tetK*. Connecting lines within the read are omitted for reasons of clarity. Genes *sdrC*, *tetK*, *sdrD* and *sdrE* are indicated. Source data are provided as a Source Data file.

might also represent an expansible/contractible locus. We integrated the *tetK* cassette between *sdrD* and *sdrE* and performed similar experiments to those described above (Fig. 4a, b). In untreated cultures, strains harboring putative amplifications of *sdrD* could be isolated from 2/6, 2/6, and 3/6 parallel cultures at day one, day two, and day three, respectively. In contrast, when CIP was incorporated, 3/6, 4/6, and 2/6 cultures harbored *sdrD* copy number variants. Copy numbers ranged from 2 to 9 copies. Again, the highest copy number variation was apparent after 2 days in the presence of CIP. To confirm the copy number of 5 ± 0.2 as quantified by qPCR for strain T17 (Fig. 4c), we performed MinION sequencing. This identified the presence of 6 tandem repeat copies but also revealed the presence of a heterogeneous T17 population, with some cells harboring >8 copies (Fig. 4d).

Altogether our results support the idea that the repetitive loci identified by the bioinformatic screen are subject to gene amplification processes and that the *csa1* and the *sdrCDE* loci

represent expandable and contractible genetic loci. Stochastic events without selection of protein function create copy number variants and generate heterogeneity even within a clonal population.

**Gene dosage correlates with protein expression levels**. We detected *csa1* amplification levels reaching up to 200 gene copies suggesting that protein expression levels of these strains are increased. We created an isogenic Δ*csa1* deletion mutant lacking the entire *csa1ABCD* locus as a control and measured Csa1 protein amounts in membrane extracts of different copy number variants using LI-COR infrared Western blotting. The four proteins encoded in the *csa1* locus are highly similar at the amino acid level. Mouse antiserum directed against Csa1C cross-reacted with the other Csa1 proteins and was used to detect all the Cas1 proteins simultaneously. The Δ*csa1* strain (0 copy number) had weakly immune-cross reactive proteins, most likely due to the

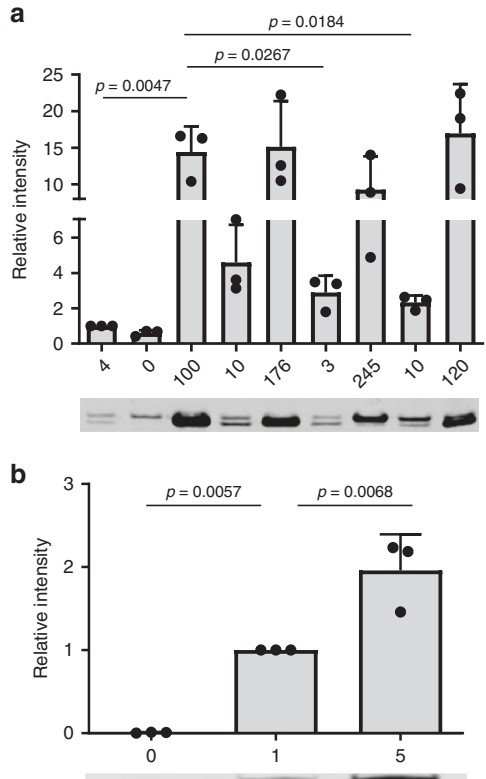

**Fig. 5 Influence of copy number variation on protein expression levels.**
**a** *csa1* copy number variants were grown to stationary phase, cell membrane fractions were isolated and analyzed by SDS-page and Western blot. Csa1 proteins were detected using α-Csa1 mouse serum, followed by goat α-mouse secondary antibody coupled to IRDye800 (LI-COR). Protein amount within bands was quantified using LI-COR infrared technology. The fluorescent signal for the WT strain harboring four copies of *csa1* was set to one in each experiment and the intensity of the copy number variants was expressed in relation to this value. Shown is a representative blot as well as the mean and SEM of three independent experiments. Statistical analysis was performed using one-way ANOVA ($F = 9{,}434$; $DF = 26$) followed by Bonferroni's multiple comparisons test. Source data are provided as a Source Data file. **b** *sdrD* copy number variants were grown to stationary phase, cell wall-fractions were isolated and analyzed by SDS-page and Western blot analysis. SdrD was detected using α-SdrD rabbit serum, followed by goat α-rabbit secondary antibody coupled to IRDye800 (LI-COR). Protein amount within bands was quantified using LI-COR infrared technology. The fluorescent signal for the WT strain harboring a single copy of *sdrD* was set to 1 in each experiment and the intensity of the copy number variants was expressed in relation to this value. Shown is a representative blot as well as the mean and SEM of three independent experiments. Statistical analysis was performed using one-way ANOVA ($F = 45{,}49$; $DF = 8$) followed by Bonferroni's multiple comparisons test. Source data are provided as a Source Data file.

presence of common epitopes in lipoproteins encoded within the other three *lpp* loci (Fig. 5a, Supplementary Fig. 2). However, compared to the WT strain harboring four copies, the signal intensity corresponding to Csa1 proteins increased in a gene-dosage dependent manner from ~2 to ~20 fold (Fig. 5a). This confirmed the gene dosage effect of *csa1* amplification. However, copy numbers higher than 120 did not result in a further increase in protein expression levels (Fig. 5a).

Similarly, *sdrD* copy number variants showed gene dosage-dependent amounts of SdrD within the cell wall, with five gene

copies increasing the amount of protein ~2 fold compared to a single gene copy (Fig. 5b).

**Amplification of *csa1* perturbs cytokine responses.** Lipoproteins expressed by *S. aureus* are the most important Microbial Associated Molecular Patterns (MAMPs)[23,24]. They are shed from the bacterial cell surface in a surfactant-dependent manner[25] and are recognized by toll-like receptor 2 (TLR2) on mammalian immune cells. The binding of lipoproteins to TLR2 activates a signaling cascade that culminates in the expression of cytokines and chemokines[24]. Therefore, we investigated whether *csa1* amplification alters the immunostimulatory capacity of the bacterial supernatants, in which the Csa1 molecules are shed. We exposed TLR2 expressing Human Embryonic Kidney cells (HEK-hTLR2) to culture supernatants of *csa1* copy number variants and found that levels of secreted IL-8 correlated with *csa1* gene dosage (Fig. 6a). Interestingly, we observed spontaneous segregation events in several lineages, whereby one of the two independent cultures used for immunostimulation had undergone a drastic reduction in copy number (e.g. E26A and E26B in Fig. 6a). The culture supernatant showed an accordingly reduced immunostimulatory capacity. This confirmed that the observed phenotypes were caused by *csa1* amplifications and were not due to secondary mutations. However, this phenomenon made replication of phenotypes for individual lineages difficult. Therefore, we grouped our samples according to the *csa1* copy number measured on the day of the experiment. This showed that an increase from four to up to 30 copies had no detectable effects on HEK-hTLR2 cells, whereas an increase from 90 to 200 copies enhanced IL-8 secretion by two- to threefold (Fig. 6b). In contrast, amplification of *csa1(FS)* locus did not increase IL-8 secretion of HEK-hTLR2 cells confirming that the observed phenotype was due to amplification induced Csa1 overexpression and not to cryptic secondary effects of the amplification (Supplementary Fig. 3c).

We hypothesized that the presence of four *lpp* loci in USA300 might mask the effects of *csa1* copy number variation especially in variants harboring rather small amplifications (up to 30 copies). Therefore, we created a triple mutant deficient in *lpl*, *lpp3*, and *lpp4* followed by the isolation of *csa1* copy number variants. We stimulated the human macrophages cell line HL60 as well as primary human polymorphonuclear leukocytes (PMNs) with culture supernatants of *csa1* copy number variants in the Δ*lpllpp3lpp4* background. Similar to HEK-hTLR2 cells, IL-8 secretion by HL60 cells and PMNs increased in a gene dosage-dependent manner. Interestingly, these experiments also allowed detection of gene dosage effects associated with ~30 copies (Fig. 6c, d).

**Gene copy number variation is created during infection.** During invasive infection pathogens face hostile host factors that undermine bacterial cellular integrity, amongst these reactive oxygen species (ROS). As ROS are known to induce DNA damage[26,27], we speculated that this might create traceable GDAs during proliferation within the organs of animals. To test this, we infected mice with a *csa1::tetK* low copy number variant and determined the *csa1* number variation within this input culture as described above. Mice were sacrificed one day post infection and the *csa1* copy number of highly Tc-resistant strains isolated from the kidneys of infected animals was determined (Fig. 7a). While copy numbers exceeding 8 copies were not detected in the cultures used for infection, we identified strains with >8 copies in 5 out of 6 infected mice. The distribution of copy numbers varied between mice, suggesting that heterogeneous populations arose independently in each mouse and generated population profiles unique to each animal. Interestingly, we found that the overall

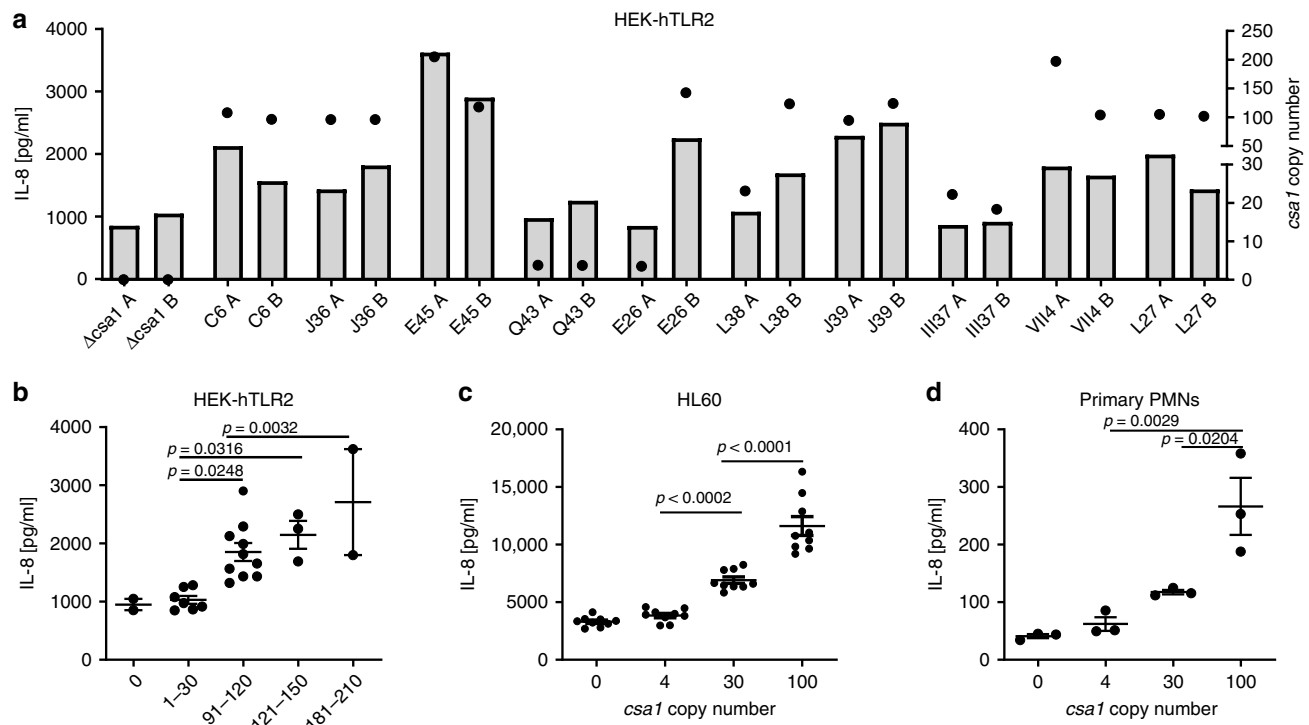

**Fig. 6 Effects of *csa1* amplification on the immunostimulatory capacity of strains. a** Confluent lawns of HEK-hTLR2 cells were stimulated for 18 h with 0.5% culture filtrates of *csa1* copy number variants grown to stationary phase. IL-8 protein levels within the supernatants were quantified by ELISA (R&D Systems). IL-8 amounts are displayed as bars referring to the left *Y*-axis, data derived from a single stimulation experiment are shown. For each lineage, two parallel bacterial cultures (labeled with A and B) were used for stimulation and the *csa1* copy number of each independent culture determined by qPCR (displayed as filled circle referring to the right *Y*-axis). Source data are provided as a Source Data file. **b** IL-8 protein levels shown in (**a**) are expressed in function of the *csa1* copy number of the stimulating strain. Sample sizes were the following: 0 - $n = 2$; 1–30 - $n = 7$; 91–121 - $n = 10$; 121–151 - $n = 3$; 181–210 - $n = 2$. Mean and SEM is shown, statistical analysis was performed using one-way ANOVA ($F = 7,710$; DF=23) followed by Bonferroni's multiple comparison test. Source data are provided as a Source Data file. **c** $5 \times 10^5$ HL60 cells were stimulated for 5 h with 1.5% culture filtrates of *csa1* copy number variants ($\Delta lpl1lpl2lpl3$ background) grown to stationary phase. IL-8 protein levels within the supernatants were quantified using ELISA (R&D Systems). Data represent three independent supernatants of each copy number variant used in three independent stimulations ($n = 9$ in each group). Mean and SEM are shown. Statistical analysis was performed using one-way ANOVA ($F = 3,632$; DF = 38) followed by Bonferroni's multiple comparison test. Source data are provided as a Source Data file. **d** Polymorphonuclear cells (PMNs) were isolated from fresh blood of healthy human volunteers. $5 \times 10^5$ PMNs were stimulated for 5 h with 1.5% culture filtrates of *csa1* copy number variants ($\Delta lpl\Delta lpp3\Delta lpp4$ background) grown to stationary phase. IL-8 protein levels within the supernatants were quantified using by ELISA (R&D Systems). Data represent three independent supernatants of each copy number variant used on PMNs of a single donor. Mean and SEM are shown. Statistical analysis was performed using one-way ANOVA ($F = 15,74$; DF = 11) followed by Bonferroni's multiple comparison test. The experiment was repeated thrice with PMNs from different donors with similar results. Source data are provided as a Source Data file.

frequency of amplifications recovered from mice was ~1000 fold higher than the frequency of amplification recovered from any in vitro passaging experiment described above (Fig. 7d). These data suggest that the *csa1* copy number variation created during infection might be favored by host selective pressures. To assess whether amplification of *csa1* in vivo depended on functional protein expression or whether it represented stochastic, unselected variation created in the environment during infection, we repeated the infection experiment using the *S. aureus* strain carrying the inactivated *csa1* locus (*csa1*(FS)::tetK) (Fig. 7b, c). Interestingly, only very few high Tc-resistant clones were isolated from infected animals 24 h post infection and none of them displayed an increased *csa1*(FS) copy number (Fig. 7b). Even when mice were sacrificed 48 h post infection and the *csa1*(FS) copy number of highly Tc-resistant strains was analyzed, we did not detect extensive heterogeneity within the *csa1*(FS)::tetK population. Indeed, only a single strain that was recovered from the kidneys of a single infected mouse carried an amplification (Fig. 7c). This strongly suggests that heterogeneity of the intact *csa1* locus was selected in vivo by a yet undefined mechanism.

The increased immunostimulatory capacity of high copy number variants observed in cell culture led us to speculate that copy number variation might influence the severity of the disease caused by the strains. To test this, we infected mice either with a low copy number variant (~4 copies) or a high copy number variant (~100 copies) of *csa1* genes and evaluated CFU counts within the organs of infected mice 72 h post infection (Fig. 7e). We did not observe an increased bacterial burden in mice infected with the high copy number variant, indicating that a high copy number of *csa1* did not result in hypervirulence. As such the evolutionary benefit of *csa1* amplification in vivo remains elusive.

## Discussion

Recombination-mediated gene copy number variations are known to contribute significantly to the plasticity of prokaryotic genomes[3,10,11,28]. Several studies have reported that gene amplifications can influence a variety of phenotypes ranging from antibiotic resistance[12,13] (see ref. [6] for an excellent review) to fitness advantages in the presence of unusual nutrients[29–32]. This suggests that this mechanism is highly relevant during adaption

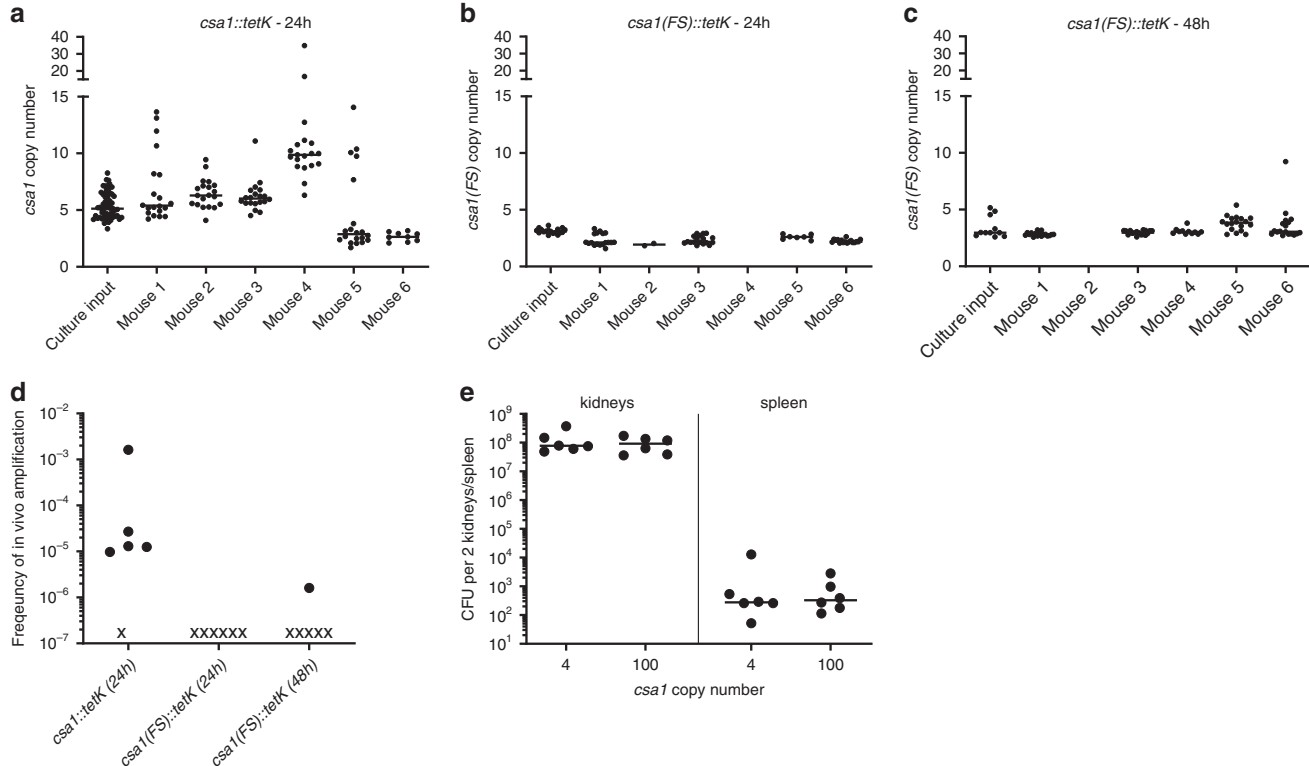

**Fig. 7 Copy number diversification in vivo. a–c** Six-week old female C57BL/6 mice were challenged with *S. aureus* live bacteria that carried either a functional *csa1::tetK* locus (**a**) or an inactivated *csa1(FS)::tetK* locus (**b**, **c**). Mice were sacrificed 24 h or 48 h post infection as indicated. High Tc resistant clones arising from the input culture and from mouse kidneys were enumerated and *csa1* (**a**) or *csa1(FS)* (**b**, **c**) gene copy number was determined. Horizontal lines show the median. Up to 19 arising clones were screened. No $Tc_{20}$ resistant clones were recovered from mouse 4 and mouse 2 in (**b**) and (**c**), respectively. Source data are provided as a Source Data file. **d** Frequency of amplification in mouse organs. qPCR analysis was used to calculate the frequency of amplification within each mouse (number of $Tc_{20}$-resistant clones that showed at least a 2 fold increase in *csa1* or *csa1(FS)* copy number compared to the parental strain within the total population of living cells recovered from each mouse) (dark-filled dots). x indicates a mouse in which amplification was not detected. Six mice were used in each group. Source data are provided as a Source Data file. **e** C57BL/6 mice were infected either with a low copy number variant (~4 copies) or a high copy number variant (~100 copies) of *csa1* genes. Mice were sacrificed 72 h post infection and CFUs within the kidneys and spleen were enumerated. Horizontal lines show the median. Statistical analysis was performed using two-tailed Mann–Whitney test, but no significant differences were found. Six mice were used in each group. Source data are provided as a Source Data file.

to stressful environments. In line with this, gene amplification is known to be important for pathogens. Poxvirus host adaption is driven by gene amplification[33]. The eukaryotic pathogen *Candida albicans* develops amplifications within its host[34] and the prokaryotes *Hemophilus influenza* and *Vibrio cholerae* can increase virulence by gene amplification[35–39].

Accordingly, one can speculate that the analysis of copy number variations in populations under constraint may pinpoint regions under evolutionary selection, which might therefore be interesting targets for experimental investigation. However, even in the age of NGS, GDAs are rarely described in environmental or clinical isolates. There might be several reasons for this. Firstly, the intrinsic instability of gene amplification arrays might result in rapid segregation as soon as an isolate is removed from its natural habitat and any selective pressure stabilizing the GDA is lifted. During isolation and culturing a GDA might therefore be lost. Secondly, current NGS strategies and associated bioinformatic analysis make detection of GDAs difficult. Generally, the individual gene copies within a GDA array are identical making the detection of GDAs using short read sequencing techniques such as Illumina challenging. However, GDAs are indicated in NGS datasets by an increased scaffolding of reads[40,41]. NGS scaffolding is rarely uniform over the entire length of the chromosome. Additionally, increased coverage might also be caused by discrete additional copies of the gene on plasmids or on distant

chromosomal locations. This makes secondary validation of presumed tandem amplifications essential. However, this is rarely within the scope of NGS projects. In our experiments copy number variation as suggested by scaffolding was confirmed by MinION sequencing demonstrating the potential of the approach.

GDAs in *S. aureus* isolates are poorly described and the known examples are only associated with increased resistance to antibiotics[42,43] or natural competence[44]. We speculated that gene copy number variations might be a prominent but neglected phenomenon. Indeed, we found the depth of NGS coverage to vary between closely related clinical *S. aureus* isolates, indicating certain genomic areas showing plasticity, especially in tandem arrays of genes with homology over the entire length of the CDSs such as *csa1*, the serine protease encoding *spl* genes[45] or the array of superantigen-like toxins (*ssl*) genes[46]. However, coverage variation of gene arrays possessing defined repetitive domains such as *sdrCDE* or of single genes with repetitive motifs such as *sasG* was also observed. These clusters can be regarded as direct evidence of the model of "innovation, amplification and divergence" which suggests that GDAs increase gene dosage of genes under selection. Divergence of gene copies by mutation creates functional diversity which is subsequently retained[47–49]. Along this line, the repetitive arrays of *S. aureus* are generally thought to encode functionally distinct proteins that are stabilized and retained within the population[50,51]. However, it has been

described that the number of repeats within these arrays can differ among *S. aureus* isolates from different lineages[52,53], suggesting ongoing diversification. Our analysis shows that even closely related clinical isolates can differ within the number of repeats and our experimental approaches show that copy number variation is created during bacterial growth in vitro and in vivo. This indicates that the loci are not as stable as generally anticipated and might suggest a long-lasting selective pressure acting on these arrays.

The scaffolding analysis identified a number of core genome loci that are frequently associated with deletions but not amplifications (e.g. *sdrCDE*). This finding can be explained by RecA-promoted recombination between sister chromatids during chromosome replication. One consequence of such recombination is one daughter cell with a duplication and another with a deletion of the DNA fragment. Duplications will be intrinsically unstable in following generations while deletions will be fixed in the progeny even if they are not beneficial to the lineage. We speculate that the loci that show frequent deletions in the collection of clinical isolates might undergo GDAs but amplifications might be lost during isolation and cultivation of the strains.

We investigated the plasticity of *csa1* and *sdrCDE* arrays in changing environments and found both loci to expand and contract, creating heterogeneous populations. Amplification in vitro was independent of protein function suggesting that stochastic recombination events drove the creation of gene copy number variation. The gyrase inhibitor ciprofloxacin, which causes DNA damage and stimulates the SOS response and RecA activity, increased diversification, supporting the idea of RecA-mediated recombination at the origin of the observed heterogeneity. Strikingly, bacterial populations recovered from infected animals exhibited *csa1* amplification only when the locus was intact. In vivo amplification of the inactivated *csa1(FS)* locus was not observed, suggesting that the in vivo function of the Csa1 confers a selective advantage and might impact host–pathogen interaction. In addition, the intact *csa1* locus amplified with a ~1000-fold increased frequency in vivo compared to any in vitro experiment. However, these datasets need to be compared with care. The bacterial generation times in vivo are unclear and bottlenecks will stochastically drive population structure during infection[54]. Nevertheless, our data suggest that heterogeneity is created in vivo and its frequency most likely enhanced by host immune pressures such as reactive oxygen species damaging the bacterial nucleic acids[55].

The biological roles of *csa1* are unclear and we therefore investigated whether amplification of the array affects immune relevant phenotypes. Bacterial lipoproteins are ligands of Toll-like Receptor 2, which recognizes diacyl and triacyl lipoproteins in combination with TLR1 and TLR6, respectively[56]. Interestingly, *S. aureus* USA300 possesses four clusters encoding Lpps (Supplementary Fig. 2), all of which are located close to the origin of replication resulting in an increase in average gene copy number due to bidirectional replication of the chromosome. The locus with the highest *lpp* copy number (*lpl – 10* genes in USA300 LAC) is located within the pathogenicity island vSaα and is important for virulence[19,24,57,58]. It was suggested that the immunostimulatory effects observed in vitro are outbalanced by the biological activities of the proteins[19]. Similar, it was shown that the expression of the *lpp4* cluster is upregulated by β-lactam antibiotics resulting in increased immune stimulation and pathogenicity[59]. Therefore, we assumed that *csa1* amplification might perturb the immune response. We found that HEK-hTLR2 cells as well as HL60 macrophages and primary human PMNs reacted with a gene dosage-dependent increase in IL-8 secretion when stimulated with supernatants of *csa1* copy number variants. Quantitative differences in the expression and release of TLR2

ligands have been shown to be crucial during *S. aureus* pathogenesis as they allow modulating host immune responses[25,60]. Highly invasive strains such as USA300 do frequently show increased TLR2 activation[60]. However, strains harboring ~100 copies of *cas1* were not hypervirulent in our infection experiments, making the benefit of in vivo *csa1* amplification ambiguous. In addition, only moderate copy number of 8–30 copies were detected in organs of infected mice. It can therefore be speculated that gene dosage-dependent expression levels might be under constant selection with benefits of amplification being outweighed by disadvantages at a certain level of amplification.

Csa1 proteins are known to be immunogenic and were proposed as vaccine candidates[18]. Additionally, recombination between *lpp* genes has been proposed to cause phase variation by creating chimeric surface-located proteins[52]. Recombination and amplification of *csa1* might therefore prevent recognition by antibodies. However, further experiments are needed to confirm this hypothesis.

We also found the *sdrCDE* locus to be expandable and contractible. This locus seems indeed to fulfill all criteria for the "Innovation, amplification and divergence" model. The DNA encoding the C-terminal SD-stalk in *sdrC*, *sdrD*, *sdrE* is highly similar. In contrast the regions encoding the N-terminal ligand-binding domains of these cell wall-anchored proteins are divergent. SdrC binds neurexin[61]. SdrD binds desmoglein 10[62] and facilitates adherence to desquamated nasal epithelial cells[63]. SdrE (Bbp) binds complement factor H[64] and bone sialoprotein[64]. Our observations support the idea that the genes were created by ancient GDA and mutations allowed functional diversification, in this case facilitating recognition of different host molecules. Our results indicate that this mechanism is still acting on this array. We did not observe amplification of *sdrD* to improve adherence to immobilized desmoglein 10. Therefore, it is tempting to speculate that the SdrCDE-encoded proteins mediate weak adherence to yet unknown ligands and that these interactions can be strengthened by increasing protein expression. However, this hypothesis remains challenging to explore. Most likely positive selection of amplifications will only occur during colonization/infection of human body sites that provide "unusual" ligands for SdrCDE. As such, only sampling strategies that strictly separate the different sites of colonization/infection, followed by NGS-scaffolding analysis, will allow a better understanding of the host factors that select for high *sdrCDE* copy number variants. The relevance of such strategies is exemplified by the work of Waller et al. which reported the frequency of gene copy number variation in *Streptococcus equi* during host adaption[41].

Finally, we also discovered that the size of repeats within an individual gene can vary significantly between closely related strains. This was observed for the cell-wall-anchored protein SasG. Such recombination harbors a risk of disrupting the open reading frame by introducing frame shift mutations. However, our analysis did not identify such mutations and we did detect SasG of different molecular weight in cell wall fractions of G5-E copy number variants. Interestingly, it has been observed before that the size of SasG varies among *S. aureus* lineages. It was also shown that very large variants of SasG promote biofilm formation by mediating Zn-dependent, intercellular interactions[53]. At the same time adherence to several host matrix molecules was reduced by long SasG variants by preventing other cell wall-anchored proteins to bind their ligands[65]. It is tempting to speculate that accordion like-expansions and contractions of SasG promote phenotype switching allowing a heterogeneous population to colonize different body surfaces.

Our experiments show that gene copy number variations are omnipresent in staphylococcal populations and can be detected by NGS analysis. Expansions and contractions of gene arrays

**Table 1 Bacterial strains used in this study.**

| Strains | Genotype | Source |
|---|---|---|
| *S. aureus* USA300 LAC | Wild type | 71 |
| *S. aureus* USA300 *cas1::tetK* | *tetK* inserted into the *csa1* locus | This study |
| *S. aureus* USA300 *cas1(FS)::tetK* | *tetK* inserted into the inactivated *csa1* locus | This study |
| *S. aureus* USA300 Δ*cas1* | Clean deletion of the *csa1ABCD* locus | This study |
| *S. aureus* USA300 Δ*sdrD* | Transposon insertion mutant derived from the Nebraska mutant library ID NE1289 | https://www.unmc.edu/pathology/csr/research/library.html |
| *S. aureus* USA300 Δ*lpl*Δ*lpp3*Δ*lpp4*Δ*cas1* | Clean deletion of all 4 *lpp* loci, | This study |
| *S. aureus* USA300:Δ*lpl*Δ*lpp3*Δ*lpp4 cas1::tetK* | Clean deletion of 3 *lpp* loci as *tetK* inserted into the *csa1* locus | This study |
| USFL037 | Clinical MRSA isolate | 15 |
| USFL085 | Clinical MRSA isolate | 15 |
| USFL086 | Clinical MRSA isolate | 15 |
| USFL091 | Clinical MRSA isolate | 15 |
| USFL118 | Clinical MRSA isolate | 15 |
| USFL162 | Clinical MRSA isolate | 15 |
| USFL165 | Clinical MRSA isolate | 15 |
| USFL190 | Clinical MRSA isolate | 15 |
| USFL202 | Clinical MRSA isolate | 15 |
| USFL225 | Clinical MRSA isolate | 15 |
| USFL234 | Clinical MRSA isolate | 15 |
| USFL275 | Clinical MRSA isolate | 15 |
| USFL308 | Clinical MRSA isolate | 15 |
| USFL311 | Clinical MRSA isolate | 15 |

occurred readily in vitro and in vivo, created heterogeneous populations and copy number variants differed in clinically relevant phenotypes. This study suggests that scaffolding analysis of NGS datasets can help identifying genomic areas under evolutionary constraint. It also suggests that scaffolding analysis of strains isolated during and after infection might pinpoint bacterial genes associated with host-adaptation, virulence or antibiotic resistance, thereby increasing the heuristic value of NGS analysis.

## Methods

**Chemicals**. If not stated otherwise, all reagents were obtained from Sigma Aldrich.

**Bioinformatic analysis of NGS data**. Whole genome sequence (WGS) data from the study by Uhlemann et al.[15] were used to look for copy number variants in the genomes of USA300 isolates. Paired-end reads were mapped against the core chromosome of the ST8 USA300 reference genome sequence FPR3757 (accession number CP000255)[16] using SMALT (www.sanger.ac.uk/science/tools/smalt-0)[41]. To identify amplifications and deletions, read coverage along the reference genome was examined using a continuous hidden Markov model with three states: 0× coverage, 1× coverage, and ≥2× coverage. Initial and transition frequencies were fitted to the data using a Baum–Welch optimization, and the most likely sequence of hidden states was calculated using the Viterbi algorithm[66].

**Bacterial strains and growth conditions**. All bacterial strains are listed in Table 1. If not specified otherwise, *S. aureus* strains were grown at 37 °C in Tryptic Soy Broth (TSB) or on Tryptic Soy Agar (TSA) (Oxoid) with shaking at 160 rpm for liquid cultures. *E. coli* strains were grown at 37 °C either in Lysogeny Broth (LB) or on Lysogeny Agar (LA) (Oxoid) with shaking at 160 rpm for liquid cultures.

**Determining copy number via MinION sequencing**. Genomic DNA was sequenced on an Oxford Nanopore Technologies MinION system (NCCT, Tübingen) with 5 k-fold mean coverage and a 10 kb mean read length. For each sample, individual raw reads containing either *tetK*, the region leading into the duplicated region (500 bp before and of the first gene in the operon), or the region following the duplicated region (500 bp of and after the last gene in the operon) were identified using NUCmer (MUMmer v3.23[67] https://anaconda.org/bioconda/mummer), run using Bioconda v4.8.0[68] (https://bioconda.github.io/). Optional arguments were used to minimize erroneous matches and allow for multiple mappings (nucmer -c 200 -maxmatch). For efficiency, only reads of at least 30 kb in length were considered. Reads identified as covering the copy region were mapped onto the entire operon (*csa1* or *sdr*) to determine gene copy numbers (nucmer -c 200, show-coords).

**Plasmid construction**. A list of plasmids used in this study is available in Table 2. The short intergenic region between *csa1B* and *csa1C* was chosen for insertion of *tetK*. A fragment containing 500 bp upstream and 500 bp downstream was synthesized (Eurofins). In this process a HindIII restriction site was introduced as well as a 5′ SacI site and a 3′ SalI site between *csa1B* and *csa1C*. The recombinant fragment was cloned blunt end into pBluescript. The resulting pBluescript:*csa* was linearized using HindIII. *tetK* was amplified from pT181 with HindIII sites at either end and cloned into pBluescript:*csa*. The resulting cassette (csa1B-tetK-csa1C) was excised from pBluescript and cloned into the thermosensitive plasmid pIMAY. For deletion of *csa1*, *lpl*, *lpp3*, and *lpp4*, 500 bp DNA fragments upstream and downstream of the genes to be deleted were amplified by PCR. A sequence overlap was integrated into the fragments to allow fusion and creating an ATG-TAA scar in the mutant allele. The 1 kb deletion fragments were created using spliced extension overlap PCR and cloned into pIMAY. All the oligonucleotides are summarized in Table 3.

For creation of the inactivated *csa1* locus (*csa1*(FS)) a recombinant locus was synthesized (Genewiz). The recombinant locus contains four point mutations that create ochre nonsense codons in triplet 3 of *csa1A*, *csa1B*, and *csa1C* as well as an opal nonsense codon in triplet 3 of *csa1D*. *tetK* was inserted into the array as described above and the *csa1*(FS)::*tetK* fragment was cloned into pIMAY. The fragment was integrated into USA300Δ*csa1* by allelic exchange[69].

**Quantitative PCR to determine the *csa1* and *sdrD* copy number**. *S. aureus* chromosomal DNA was isolated using the BioEdge chromosomal DNA isolation kit and Quick-DNA 96 Plus Kit according to the manufacturer's recommendation with an additional incubation with 1 µg/ml lysostaphin for 1 h after resuspension. Chromosomal DNA was adjusted to 100 ng/µl and a 1:10 serial dilution was used to create standard curves. qPCR primers were designed using the Primer3 software and are listed in Table 3. Primers directed against the origin of replication (*ori*) were used as the single copy reference. Primers against *csa1* were directed against a highly conserved stretch of the coding sequence to allow amplification of a fragment from all four genes (*csa1A/B/C/D*). Primer binding sites were only partly conserved in the other *lpp*-coding sequences. No amplification was detected using chromosomal DNA of the Δ*csa1* strain, ruling out amplification of *lpl*, *lpp3* and *lpp4* genes. Primers against *sdrD* amplified a 5′ fragment of *sdrD*. The relative abundance of *csa1* and *sdrD* in relation to *ori* was calculated using standard curves. qPCR was performed using the Quantstudio3 (Applied Biosystems) and the "SYBR Green Mastermix" (Applied Biosystems).

**Bacterial growth to isolate copy number variants**. *S. aureus* USA300 *csa1::tetK* was used to inoculate 20 ml TSB and incubated for 6 h at 37 °C. Cells were harvested, washed and used to inoculate 20 ml TSB at an $OD_{600} = 0.05$. When indicated, ciprofloxacin (Fluka) was added to the culture. Cultures were incubated for 24 h at 37 °C. The next day bacteria were diluted (1:20) in fresh TSB and incubated for additional 24 h at 37 °C. After incubation serial dilutions were prepared and CFUs on TSA (total counts) and on TSA containing 20 µg/ml tetracycline ($TSA_{Tc20}$, putative amplifications) were enumerated. The *csa1* copy number of

**Table 2 Plasmids used in this study.**

| Plasmids | Description | Source |
|---|---|---|
| pT181 | Staphylococcal plasmid encoding *tetK* | 72 |
| pIMAY | Thermosensitive vector for allelic exchange | 69 |
| pIMAY:*csa1::tetK* | Fragment for insertion of *tetK* into the *csa1* locus of USA300 LAC | This study |
| pIMAY:*csa1*(FS)::*tetK* | Fragment for intergration of the inactivated *csa1-tetK* locus into USA300 Δ*csa1*. | This study |
| pIMAY:Δ*csa1* | Fragment for deletion of the *csa1* locus of USA300 LAC | This study |
| pIMAY:Δ*lpl* | Fragment for deletion of the *lpl* locus of USA300 LAC | This study |
| pIMAY:Δ*lpp3* | Fragment for deletion of the *lpp3* locus of USA300 LAC | This study |
| pIMAY:Δ*lpp4* | Fragment for deletion of the *lpp4* locus of USA300 LAC | This study |

**Table 3 Oligonucleotides used in this study.**

| Name | 5′–3′ sequence | purpose |
|---|---|---|
| csa1_Sc.F | GATATTAAGACGAGTATGAAAATAGTTAG | Screening for length variation in the *csa1* locus. |
| csa1_Sc.R | ATTTTACAGCAACATATTTGAATTTC | Screening for length variation in the *csa1* locus. |
| qCsa1_F | TCCAGAGGTGCCGAGTTATT | qPCR of the *csa1* locus. |
| qCsa1_F | TTTATATCCAACTGATGAGCCTTTT | qPCR of the *csa1* locus. |
| qOri_F | TCGTGATAACGAAGGTGAAGC | qPCR origin of replication. |
| qOri_R | GGTGGTCGATCACTCGAAAT | qPCR origin of replication. |
| qSdrD_F | GCGACAACTTCAGCAAGTGA | qPCR *sdrD*. |
| qSdrD_R | TGGTGAAGCTTGCTCATCTG | qPCR *sdrD*. |
| sdr_Sc.F | GAGCAATGTTATTAATTAAAATAAGATG | Screening for length variation in the *sdrCDE* locus. |
| sdr_Sc.R | GAATAAGGATTCCATTTAACATATACAC | Screening for length variation in the *sdrCDE* locus. |
| 0293_Sc.F | GGAAATAAGTGTAGAGAATAAATTAATAG | Screening for length variation in the SAUSA300_0293...0296 locus. |
| 0296_Sc.R | TATTATTATTTGATGACAACTTTATGG | Screening for length variation in the SAUSA300_0293...0296 locus. |
| csa1KO_A | AGGGAACAAAAGCTGGGTACCACTATGATAAAAAAGTTGAAG | Construction of *csa1* deletion cassette. |
| csa1KO_B | CATCTTACAACTCTCTTCTTTTTAAAATG | Construction of *csa1* deletion cassette. |
| csa1KO_C | GAAGAGAGTTGTAAGATGTAATCATCCACACACGATTC | Construction of *csa1* deletion cassette. |
| csa1KO_D | ATAGGGCGAATTGGAGCTCCATAAGCAACTGAATCACAAG | Construction of *csa1* deletion cassette. |
| csa1_ScF | CATTTGAAACGAAAATTAATAATGG | Screening for *csa1* deletion |
| csa1_ScR | CTTTTGGTTCGAATGATATGTACGC | Screening for *csa1* deletion |
| lplKO_A | CACTAAAGGGAACAAAAGCTGGGTACCCAATATAACTTAATTCATGTTCTAAG | Construction of *lpl* deletion cassette. |
| lplKO_B | CATATAAATAATTAATTATTTTGTATATTCTC | Construction of *lpl* deletion cassette. |
| lp1KO_C | CAAAATAATTAATTATTTATATGTAGGAAGTATAAAATAGATTTAAAAG | Construction of *lpl* deletion cassette. |
| lplKO_D | ACTCACTATAGGGCGAATTGGAGCTCTAGAACTACCGCATCTCTTCCACCTA | Construction of *lpl* deletion cassette. |
| lpl_ScF | AAGTATGATCTTAAGTTGTCTTTTGTAGC | Screening for *lpl* deletion |
| lpl_ScR | TAGAAATAGGAGCTGGATTATAAACC | Screening for *lpl* deletion |
| lpp3KO_A | CACTAAAGGGAACAAAAGCTGGGTACCATATTGATGCTATTTCAATTGCAGG | Construction of *lpp3* deletion cassette. |
| lpp3KO_B | CATACATTCCCACCGTTTCTCAAAATAC | Construction of *lpp3* deletion cassette. |
| lpp3KO_C | GTATTTTGAGAAACGGTGGGAATGTATGTAATACTTATGCTGTAATTATAGAAAC | Construction of *lpp3* deletion cassette. |
| lpp3KO_D | CGACTCACTATAGGGCGAATTGGAGCTCGAAGTTAGTGCACATATTGAAGATTTAAG | Construction of *lpp3* deletion cassette. |
| lpp3_ScF | TATGTATTTGTAACGCCTATGTGGAACC | Screening for *lpp3* deletion |
| lpp3_ScR | CGATGGATGCATGACAAATATTGGG | Screening for *lpp3* deletion |
| lpp4KO_A | CACTAAAGGGAACAAAAGCTGGGTACCGGAATTAAAATGTATATTTTTGTACAG | Construction of *lpp4* deletion cassette. |
| lpp4KO_B | CATTTCACATCCCCATTTTTATTTTTG | Construction of *lpp4* deletion cassette. |
| lpp4KO_C | AAATAAAAATGGGGATGTGAAATGTGAATATCAAATAAAACCTGGTAATA | Construction of *lpp4* deletion cassette. |
| lpp4KO_D | CGACTCACTATAGGGCGAATTGGAGCTCTACAAGTCTAATATTACATGAATTTCC | Construction of *lpp4* deletion cassette. |
| lpp4_ScF | TAAAATGGTTTACTAAATCTAATAGAAC | Screening for *lpp4* deletion |
| lpp4_ScR | TTATATAAACTCTCTCGTCTCTCTCTA | Screening for *lpp4* deletion |
| Sdr_tetK_A | ACTCGAGCTCAACCAATGAGTACGG | Insertion of *tetK* into the *sdrCDE* locus. |
| Sdr_tetK_B | ATCGTAAAACGGGATCCAACATTTGTGT | Insertion of *tetK* into the *sdrCDE* locus. |
| Sdr_tetK_C | GGATCCCGTTTTACGATAAAGAAAAATAATTAAAGTATTG | Insertion of *tetK* into the *sdrCDE* locus. |
| Sdr_tetK_D | CATTTGTCGACGTTTCATTACCTTGAGA | Insertion of *tetK* into the *sdrCDE* locus. |
| Sdr_tetK_Scr.F | GATAGCGATTCAGATTCAGATGCAG | Screening of *tetK* insertion into the *sdrCDE* locus. |
| Sdr_tetK_Scr.R | CAACTTTATTTCCAGTGGTAGATTGTACAC | Screening of *tetK* insertion into the *sdrCDE* locus. |
| tetK_F | GTCAACGGGGTTTTCAATGGGGAAAGCTTCACAGAA | Amplificaton of *tetK* from pT181. |
| tetk_R | CATAACACTAACAAAACATCGCTGTTAAAGCTTTTTTATTAC | Amplificaton of *tetK* from pT181. |

highly tetracycline (Tc) resistant clones was determined by qPCR. Due to the high number of isolates to be screened, we performed the qPCR analysis once for each isolate. Copy numbers of strains used for further investigations were confirmed by repeating the qPCR measurement. To determine amplification frequency within a population, we interpreted a doubling in signal strength ($csa1/ori$) as amplification of $csa1$. The frequency of amplifications within the culture was calculated using the formula: $\text{freq} = \frac{\text{amplifcations}\left(\frac{\text{CFU}}{\text{ml}}\right)}{\text{total CFU}\left(\frac{\text{CFU}}{\text{ml}}\right)}$.

**Isolation of bacterial membranes and detection of Csa1 proteins.** Fractionation was carried out as described earlier[70] with minor modifications. Briefly, cells were grown in TSB to stationary phase and washed once with wash buffer WB (10 mM Tris-HCl pH 7, 10 mM MgCl). A 1 ml aliquot of cells adjusted to an $OD_{578} = 5$ was centrifuged ($18.000 \times g$) and resuspended in 100 μl digestion buffer (10 mM Tris-HCl pH 7, 10 mM MgCl, 500 mM sucrose, 0.3 mg/ml lysostaphin, 250 U/ml mutanolysin, 30 μl protease inhibitor cocktail (Roche – 1 complete mini tablet dissolved in 200 μl H₂O), 1 mM phenyl-methanesulfonylfluoride (PMSF). The digestion of the cell wall was carried out at 37 °C for 1 h followed by centrifugation ($3.000 \times g$ for 20 min at 4 °C). The supernatant was designated "cell wall fraction". The pellet containing the protoplasts was washed with 1 ml WB (with 500 mM sucrose) and centrifuged again as above. The protoplasts were resuspended in 200 μl buffer L (100 mM Tris-HCl pH 7, 10 mM MgCl, 100 mM NaCl, 10 μg/ml DNaseI, 100 μg/ml RNaseA). The suspension was frozen and thawed three times to ensure complete protoplast lysis and centrifuged for 30 min ($18.000 \times g$ at 4 °C). The pellet (designated "membrane fraction") was washed with 1 ml of buffer L and resuspended in 100 μl TE buffer (100 mM Tris-HCl pH 8, 1 mM EDTA). 5–15 μl of the fractions were used for analysis by SDS-PAGE. Western immunoblotting was performed using standard procedures and mouse serum directed against Csa1C (1:1000)[18]. Secondary goat α-mouse-DYLight 800 antibodies (LI-COR 926-32210) were used and fluorescence intensity was quantified using the "Odyssey ClX" Infrared technology from LI-COR.

**Isolation of cell wall fraction and detection of SdrD proteins.** The exponential cultures were inoculated from the 20 ml overnight cultures and grown from an $OD_{600} = 0.1$ to 0.9 in TSB. The cell-wall fraction was isolated as described above, 10 μl of sample loaded on a 7.5% gel and analyzed by Western immunoblotting using rabbit antiserum against SdrD (1:1000) (kind gift of T.J. Foster) and goat α-rabbit-DYLight 800 secondary antibodies (LI-COR 926-32211).

**Mouse bacteraemia model.** Animal experiments were performed in strict accordance with the European Health Law of the Federation of Laboratory Animal Science Associations. The protocol was approved by the Regierungspräsidium Tübingen (IMIT1/17). Mice were kept in 360 cm² (Type 2) individually ventilated cages (3 mice per cage) with food and water ad libidum, 12 h light, 22 °C and 53% humidity. TSB medium was inoculated with *S. aureus* strains from a pre-culture and grown to mid-exponential growth phase (2–3 h). Bacteria were harvested, washed and diluted in sterile PBS. CFUs were determined by diluting and plating on TSA and $TSA_{Tc20}$. Six-week old female C57BL/6 wild-type mice purchased from Envigo were challenged with $1 \times 10^7$ live bacteria in 0.2 ml PBS injected into the tail vein. Survival and disease progression was monitored for up to 72 h and viable counts of bacteria in the organs were enumerated on TSA plates as well as on $TSA_{Tc20}$. $csa1$ copy number of up to 20 Tc resistant colonies was determined by qPCR as described above.

**Stimulation of cell lines.** Stimulation of the cell lines was performed with the sterile filtered bacterial culture supernatants in TSB, diluted to a final concentration of 0.5% in a suitable medium with incubation at 37 °C and 5% $CO_2$. HEK cells stably transfected with the human TLR2 gene were purchased from InvivoGen (France) and cultivated in 75 cm² culture flasks using 20 ml of growth medium (Dulbecco's modified eagle medium (DMEM), 10% fetal calf serum (FCS), 100 μg ml⁻¹ normocin, and 10 μg ml⁻¹ blasticidin). HEK cells were cultivated in DMEM, 10% FCS, 20 mM L-glutamine, and 1000 u ml⁻¹ penicillin/streptomycin. $0.3 \times 10^6$ cells were seeded into 24-well cell culture plates and cultivated until confluence was reached. Growth medium was then replaced by medium without FCS containing appropriately diluted stimuli and the plate was incubated for 18 h. HL60 cells were purchased from the Leibniz Institute DSMZ—German Collection of Microorganisms and Cell Cultures (Germany) and cultivated in a 75 cm² culture flasks using 20 ml of growth medium (very low endotoxin-Roswell Park Memorial Institute Medium (RPMI) 1640, 10% FCS, 2 mM GlutaMax, 100 u ml⁻¹ penicillin/streptomycin, 10 mM 4-(2-hydroxyethyl)-1-piperazineethanesulfonic acid (HEPES)). $5 \times 10^5$ cells were seeded into U shape bottom 96-well cell culture plates and immediately stimulated with the diluted culture filtrates for 18 h. Diluted culture filtrates exerted no toxicity towards HEK and HL60 cells as analyzed with the Cytotoxicity Detection Kit (Roche Applied Sciences). No stimulatory activity was detected in non-inoculated media at corresponding dilutions. After stimulation supernatants were collected by centrifugation for 10 min at $250 \times g$ and stored at −20 °C before use. Cytokines were diluted and measured using ELISA kits (R&D Systems) according to the manufacturer's instructions.

**Stimulation of primary cells.** Human PMNs were isolated from venous blood of healthy volunteers in accordance with protocols approved by the Institutional Review Board for Human Subjects at the University of Tübingen. Informed written consent was obtained from all volunteers. PMNs were isolated by standard Ficoll/Histopaque gradient centrifugation and stimulated with diluted bacterial culture filtrates (final concentration 0.25%) in 96-well U shaped bottom plates. $5 \times 10^5$ PMNs were seeded in cell culture medium (very low endotoxin RPMI 1640, 2 mM sodium pyruvate, 2 mM L-glutamine, 100 u ml⁻¹ penicillin/streptomycin, 10 mM 4-(2-hydroxyethyl)-1-piperazineethanesulfonic acid (HEPES)) and incubated for 5 h at 37 °C in 5% $CO_2$. Cytotoxicity and cytokines were measured as described above.

**RecA expression.** *S. aureus* USA300 LAC was used to inoculate 20 ml TSB and incubated for 18 h at 37 °C. 20 ml broth samples were inoculated to an $OD_{600} = 0.1$ and 0.4–2 μg/ml ciprofloxacin was added. The $OD_{600}$ was measured each hour for 8 h and then after 24 h. At $OD_{600} = 0.9$ 1 ml of each culture was harvested, the cells were lysed using glass beads. Cell debris was pelleted ($17.000 \times g$, 10 min) and the supernatant collected. Protein concentrations of the crude extracts were determined by Bradford and 1 μg of total protein was used for SDS-PAGE and Western immunoblotting using standard procedures. Filters were incubated with rabbit α-RecA antibody (1:3000) (Abcam, ab63797)[18] and goat α-rabbit IgG-DYLight 800 (LI-COR 926-32211) and fluorescence intensity was quantified using the "Odyssey ClX" Infrared technology.

**Statistical analysis.** Statistical analysis was performed using graphpad Prism. The used tests as well as the number of replicates for each experiment are indicated in the respective figure legends.

**Reporting summary.** Further information on research design is available in the Nature Research Reporting Summary linked to this article.

## Data availability
MinION sequencing data are deposited in the NCBI BioProject database under the accession number PRJNA601323w. The USA300 datasets analyzed here were previously deposited in the European Nucleotide archive under accession number PRJEB2870 (ref. [15]). All data obtained or analyzed in this study underlying the figures in this manuscript are available in Supplementary Data 1 or in the Source Data file. Source data are provided with this paper.

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

## Acknowledgements

The authors thank Timothy J. Foster for helpful discussion, editing the manuscript and for providing αSdrD and αSasG antiserum. We thank Libera Lo Presti for critically reading and editing this manuscript. We thank Andreas Peschel for helpful discussion. We thank GlaxoSmithKline (GSK) for providing αCsa1 antiserum. We acknowledge the funding support of the Research Executive Agency to SH (https://erc.europa.eu/). This project has received funding from the European Union's Horizon 2020 research and innovation program under the Marie Sklodowska-Curie grant agreement No "GA 655978". We acknowledge the support of the University of Tübingen (EKUT) as well as the Ministry for Science and Art Baden-Württemberg for support via the RiSC initiative to SH. SH was supported by infrastructural funding from the Deutsche Forschungsgemeinschaft (DFG), Cluster of Excellence EXC 2124 Controlling Microbes to Fight Infections. MTGH was supported by Chief Scientists Office (Reference: SIRN10). None of the funding bodies was involved in the design of the study, the performance of experiments, data evaluation, writing of the manuscript or the decision about submission.

## Author contributions

D.B. and A.J. performed, designed and evaluated experiments. J.P. evaluated MinION sequencing data. M.T.G.H. performed bioinformatic data curation for NGS scaffold analysis and provided assistance for data interpretation. S.H. designed the study, performed bioinformatic evaluation as well as experiments and wrote the manuscript.

## Competing interests

The authors declare no competing interests.
