## [Peer Review File · Nature Communications]

Reviewers' comments:

Reviewer #1 (Remarks to the Author):

The study by Belikova et al. investigates the biological relevance of scalable gene amplification, which is a relevant part of the evolutionary ecology of bacteria. Starting with bioinformatics the authors identify several putative genomic regions that might be under selection for gene amplification in published *Staphylococcus aureus* genomes. They then examine the biological relevance of gene amplifications that span *csa1*. The authors show that amplifications of this region readily occur *in vitro* and *in vivo* and can be scaled by the antibiotics tetracycline/ciprofloxacin. As bacterial lipoproteins (e.g. Csa) are key players in pathogen-host-interaction (recognized by TLR2), the authors also suggest that *csa* copy number variants differ in their immuno-stimulatory capacity and that this might be a mechanism to modulate the host response.

While the results are interesting, the study is not as novel as claimed by the authors and several of the experiments lack essential negative controls.

Major Points:

1. The findings of the study are not quite as novel as stated by the authors (L. 55-57). There are numerous studies which show that antibiotics can select for increased copy number (see e.g. ref. 5 in manuscript). A more recent example comes from Laehnemann et al 2014 who showed that doxycycline and erythromycin antibiotics stimulates a scalable increase in the copy number of the genomic region that spans the multidrug-efflux pump *acrAB-tolC*. There is also a nice example (Elde et al Cell 2012) of how Poxviruses "use" gene amplification to counteract host defences.

2. The experiments reported show correlations between *csa* copy number and immune response (Fig. 4), and an altered copy number distribution of *csa* in mice (Fig. 5) that is supposedly reflecting adaptation. These experiments lack essential negative controls to separate effects of altered Csa levels from (i) indirect effects caused by gene amplification/recombination of the *csa* region or (ii) stochastic effects:

a. For all experiments a control is needed where the *csa* gene is inactivated (e.g. by a small in-frame deletion).

b. Likewise, a control is needed for the exp. in Fig. 4 to achieve altered Csa levels in another way than by gene amplification, i.e. by expression from an inducible promoter.

c. Finally, for the animal experiments (Fig. 5) a control is needed to determine the copy number variation of regions that are not under selection. This control is also required to separate technical from biological variation for the analyses in Fig. 1C and Fig. 2.

3. Regarding the animal experiments in Fig. 5, could not this result simply be the result of a stochastic founder effect? Note that mice 5-6 have lower average copy number than input and mice 2-3 are not that different from input. To me this looks a bit like the outcome of a Luria-Delbruck experiment.

Minor Points:

1. Fig 1C. The PCR fragment for USFL isolate 165 is not convincing. As this isolate happens to be the one with the highest copy number according to qPCR, an experimental validation of these results would be helpful.

2. The lines in Fig. 2C are not labelled, thus the strength of selection in the evolutionary experiments cannot be assessed.

Reviewer #2 (Remarks to the Author):

Belikova et al. investigated the gene copy number variations in natural populations of bacteria which causes genotypic and phenotypic heterogeneity in clonal populations of *S. aureus* USA300. The study shows the expansion and also the contraction of gene arrays occur readily in vitro and in vivo with a high various alteration up to more than 100 gene copies in vitro and 30 copies in vivo. This is an interesting topic which explains the tremendous ability of prokaryotes to adapt to changing environments. In most cases selection of point mutations contribute to adaptation; for example respiration defective mutants that lead to small colony variants, SCVs have an advantage by causing persistent infections. Here it was shown that also gene duplication or even multiplication occurred. With the investigation of some representative host colonization and virulence factors such as surface anchored molecule SdrD, the Spl serine proteases and lipoprotein gene *sca1*, the authors pinpoint the gene amplification occurrence during colonization and infection which is controlled by the balance of the bacterial benefits and the disadvantage to the host. However, the authors should consider the following points:

Major points:

1. As far as can be seen there was only one pathogen investigated, *S. aureus*, and even not various *S. aureus* strains, but only the *S. aureus* USA300 lineage. This should be then also stated in the title.
2. Lpl3C (SAUSA300_2424) and Lpl2 (SAUSA300_0205) are not lipoprotein genes as they don't contain the right signal peptide with lipobox which is necessary for Lgt and Lsp function (Shahmirzadi 2016 <https://www.frontiersin.org/articles/10.3389/fmicb.2016.01404/full>). The DOLOP is not the up to date program to search for the bacterial lipoproteins (Line 110-111). The author should correct them.
3. *csa1A-D*, *lpl1-9* and tandem *lpp* (SAUZA300_2429, SAUSA300_2430) share the highly conserved core region 38 aa (Shahmirzadi 2016). However, the author mentioned in Line 394-396 that primers against *csa1* were directed against a highly conserved stretch of the coding sequence to allow amplification of a fragment from all four genes (*csa1A/B/C/D*). The authors should make it clear how they can detect the specific genes for *csa* cluster but other lipoproteins, such as *lpl*.
4. USA300 has 3 different tandem lipoprotein clusters: *csa* cluster (*Csa1A/B/C/D*), *lpl* in *vsaa* island, the rest SAUSA300_2429-2430 should be named differently (such as tandem *lpp* cluster) to distinguish them.

Minor points:

1. Line 202-203: instead of REF 20, the author should rather mention the Ref 21 and REF (Stoll 2005) <https://www.ncbi.nlm.nih.gov/pubmed/15784587> .
2. Line 230: ..high *csa1* high copy....
3. Line 299: instead of REF 21, the author would rather mention the Ref (Nguyen et al. 2017, <https://www.nature.com/articles/s41467-017-02234-4>).
4. Line 406: there is a typo
5. Fig1A III part: there should be 4 genes (*csa1A* to *csa1D*) but 5 genes are shown.
6. Fig 2C lacking the explanation of the CIP concentration for each the growth curve.
7. Fig 2E: lacking the title for Y-axis.

Reviewer #3 (Remarks to the Author):

The title refers to gene copy number variation which is usually a term for multiple numbers of copies of the same gene in a cell, for example those on plasmids or parts of the genome that have replicated to multiple copies. In the text, an alternative is also discussed – gene duplication and amplification (GDA), where a second copy of a gene is introduced into the chromosome downstream of the original copy. These are two different concepts and the terminology used in the title and throughout the manuscript was therefore confusing and conflicting to this reviewer. Are we discussing copies or duplicates? What does the term amplification refer to here?

The first results section identifying gene copy number in staphylococcal chromosomes is not novel. The concept here described as GDA is well known in *S. aureus* chromosomes for a range of genes, particularly in the pathogenicity islands (examples Jarraud et al., *J Immunol* 2001; Al Shangiti et al., *Infect Immun* 2004). Although these genes were likely generated by duplication, each individual gene has a unique function, eg. Ssls, This would argue that many GDAs are stable, and functional. Another similar and well described duplication is in the repetitive binding units within genes, such as *spaA* used for *S. aureus* typing where one or two duplications arising in a short period are seen (Boye & Westh, *FEMS Micro Lett* 2011). In all cases, the maximum number of duplicates is around 20 and does not normally arise from bacterial subculture. This context would be useful for the paper. The authors should also consider discussion of the limitation of using short read sequencing to estimate gene copy number.

The novel Results begin by focusing on a single gene, *csa1* and showing that GDA duplicates can be seen in different isolates. Figure 1 justifies the use of qPCR as a method for identifying the numbers of duplicate *csa1* genes in isolates with 1-4 duplicates.

Figure 2a. shows evidence of *csa1* duplication after standard growth in the laboratory. qPCR amplification suggests over 100 duplications. But I do not see the evidence that *csa1* gene has duplicated 100 times. SMRT or minIon sequencing showing the duplications would be useful. Without this evidence, it suggests that qPCR is not be a robust method for detecting large numbers of duplications.

Since the authors propose multiple genes are duplicated, and this occurs in normal growth conditions, why are whole genome sequencing using SMRT technology not identifying isolates with 100 or more duplications of multiple genes? Could the qPCR method be amplifying mRNA generating artificially high counts in the experiments? Or that some regions of the chromosomal DNA are replicating at higher frequency?

In Figure 2b the y-axis shows very low numbers but it is not clear from the legend or the methods how this experiment was performed. What is being measured?

From here on there is such a doubt raised about the validity of the qPCR to detect gene duplication that it is not clear how the rest of the data should be interpreted.

Therefore, while the concept of this paper is interesting, the results are preliminary without the necessary robust evidence that the genes described are in fact duplicating.

Response to Reviewer Comments:

Reviewers' comments:

Reviewer #1 (Remarks to the Author):

The study by Belikova et al. investigates the biological relevance of scalable gene amplification, which is a relevant part of the evolutionary ecology of bacteria. Starting with bioinformatics the authors identify several putative genomic regions that might be under selection for gene amplification in published *Staphylococcus aureus* genomes. They then examine the biological relevance of gene amplifications that span *csa1*. The authors show that amplifications of this region readily occur in vitro and in vivo and can be scaled by the antibiotics tetracycline/ciprofloxacin. As bacterial lipoproteins (e.g. Csa) are key players in pathogen-host-interaction (recognized by TLR2), the authors also suggest that *csa* copy number variants differ in their immuno-stimulatory capacity and that this might be a mechanism to modulate the host response.

While the results are interesting, the study is not as novel as claimed by the authors and several of the experiments lack essential negative controls.

Major Points:

1. The findings of the study are not quite as novel as stated by the authors (L. 55-57). There are numerous studies which show that antibiotics can select for increased copy number (see e.g. ref. 5 in manuscript). A more recent example comes from Laehnemann et al 2014 who showed that doxycycline and erythromycin antibiotics stimulates a scalable increase in the copy number of the genomic region that spans the multidrug-efflux pump *acrAB-tolC*. There is also a nice example (Elde et al Cell 2012) of how Poxviruses “use” gene amplification to counteract host defences.

*We agree with the notion that gene amplifications improving antibiotic resistance are well described. This was discussed in the original manuscript and we now include the reference proposed by this reviewer in the revised paper (ln52 and 304). We also note that we did not carefully formulate the remark in Line55-57 and that we referred to loci that are not involved in promoting antibiotic resistance but show variation in response to stress. This is now expressed more carefully (line 53-60). To the best of our knowledge, environmental factors, such as antibiotic pressure or host immune defenses, that promote copy number variation in these loci have not yet been described. The novelty of our study is in the finding that many repetitive loci in *S. aureus* undergo constant amplification/deletion, thus generating heterogeneous *S. aureus* populations. This process can be favored by antibiotic-induced DNA damage but the loci themselves are not involved in promoting antibiotic resistance but rather entail altered host pathogen interactions. Gene amplifications.*

Some rare examples suggest that loci involved in host-pathogen interaction can be the target of GDAs in eukaryotic and prokaryotic pathogens, as well as in Poxviruses (pointed out by this reviewer). However, to the best of our knowledge frequency induction and effects of amplifications have not been studied in prokaryotes. We now acknowledge this and include references to Poxviruses and fungal pathogens in our discussion section (ln 306-310).

2. The experiments reported show correlations between *csa* copy number and immune response (Fig. 4), and an altered copy number distribution of *csa* in mice (Fig. 5) that is supposedly reflecting adaptation. These experiments lack essential negative controls to separate effects of altered Csa levels from (i) indirect effects caused by gene amplification/recombination of the *csa* region or (ii) stochastic effects:

a. For all experiments a control is needed where the *csa* gene is inactivated (e.g. by a small in-frame deletion).

*To address this point, we constructed an isogenic strain carrying an inactivated version of the *csa1::tetK* locus (named *csa1(FS)::tetK*), in which each of the 4 genes was inactivated by a nonsense*

mutation (Fig. S3, ln472-476). This locus is still repetitive but is unable to confer any benefit/burden associated with protein overexpression. Therefore, it allows to discriminate the effects due to altered Csa1 levels from indirect or stochastic effects of the amplification.

We now show that, like the functional *csa1::tetK* locus, also the inactive *csa1(FS)::tetK* locus undergoes gene amplification during *in vitro* passaging and this can be enhanced by ciprofloxacin treatment at day 2 (Fig S3, ln200-205). This suggests that *in vitro* generation of heterogeneity is stochastic rather than the result of selective pressure on Csa levels.

However, unlike the WT locus, the *csa1(FS)::tetK* locus was not amplified after *in vivo* evolution in mice (Fig.7, ln282-292). This suggests that the heterogeneity observed for the intact *csa1* locus after *in vivo* passage is favored by unknown selective pressures.

b. Likewise, a control is needed for the exp. in Fig. 4 to achieve altered Csa levels in another way than by gene amplification, i.e. by expression from an inducible promoter.

Expression of bacterial lipoproteins using recombinant plasmids alters the immunostimulatory capacity of strains and this is well documented in the literature [1, 2] and others.

Additionally, expression levels comparable to the 150 fold levels that we observed in our assays, e.g. by the C6 variant, cannot easily be achieved with the available plasmids.

Therefore, we do not feel that this manuscript would benefit from this additional control of vector driven expression.

However, we agree that it should be demonstrated that the observed overexpression is due to increased Csa levels is not caused by secondary-site mutations or unknown indirect effects of the amplification.

As discussed below as well, we found several times that long tandem arrays collapsed and enabled reisolation of low copy number variants from high copy number progenitors. Always associated with this collapse was the decrease of the immunostimulatory capacity, strongly suggesting that the amplification caused the observed phenotype. One example is given in the manuscript (Fig 6, E26A and E26B, ln250-253).

*Additionally, we now show that only amplifications associated with functional *csa1* genes give rise to increased immunostimulatory capacity (line257-259, Fig S3C). This demonstrates, that it is the functional overexpression of *csa1* rather than cryptic effects of the amplification responsible for the increased immunostimulatory capacity.*

c. Finally, for the animal experiments (Fig. 5) a control is needed to determine the copy number variation of regions that are not under selection. This control is also required to separate technical from biological variation for the analyses in Fig. 1C and Fig. 2.

As our mice are not receiving antibiotic treatment, we think the Reviewer is referring to “regions that are not under selection” by the immune system.

*In our eyes, the only control strain that would allow the detection of amplification of genes that are definitely not under selection is the *csa1(FS)::tetK* strain described above. This is now included and suggests that the observed heterogeneity after *in vivo* evolution was not purely stochastic but favored by selective pressures (Fig.7, ln282-292).*

With respect to controls for technical and biological variation, we note that in this context it is frequently difficult to discriminate between biological and technical variation, as especially high gene copy number variants are intrinsically unstable. Hence rapid segregation is frequently observed leading to overnight cultures possessing different copy-numbers and thereby different immunostimulatory capacity although the cultures were initially inoculated from the same agar plate.

To illustrate this, we mentioned in the original manuscript the example of E26A and E26B (Fig 6 In250-253). Massive differences in the immunostimulatory capacity of these “biological replicates” correlated with massive differences in gene copy-number that were measured in the culture at the day of the experiment. Therefore, we suggested to regard the gene copy number measured at the day of the experiment as a means for “biological replication” and grouped strains with similar copy number together to compare their immunostimulatory capacity (Fig. 6b). This approach enabled us to show strict dependence between immune-stimulation and *csa1* copy number.

In Fig 1C technical replicates (3 qPCRs from a single chromosomal DNA isolation) are shown. We also performed biological replication using three DNA samples isolated from three independent cultures for each strain and run qPCR in technical replicates. Fig. R1 shows the results of this biological replication. Minor differences among the measured copy numbers for the independent cultures were observed. Biological replicates 2 and 3 were performed about three years after Biological replicate 1

Regarding Fig 2 (now split in Fig. 2 + Fig. 4), this figure represents data from 6 independent passage experiments, i.e. “biological replicates”. Due to the number of samples (~650), technical replication of qPCR results was not performed as the technique is generally robust (see Fig 1C or 4A where technical replication was performed).

We assume that none of the genes in this assay is under selection as tetracycline is not present in the liquid culture! The events are therefore most likely stochastic. This is supported by the finding that we observed amplification *in vitro* also for the *csa1*(FS) control strain (Fig.S3).

3. Regarding the animal experiments in Fig. 5, could not this result simply be the result of a stochastic founder effect? Note that mice 5-6 have lower average copy number than input and mice 2-3 are not that different from input. To me this looks a bit like the outcome of a Luria-Delbruck experiment.

We agree, this is possible. Due to experimental limitations we cannot prove that amplification of *csa1* is selected within the host. Our infection with WT vs high *csa1* copy number rather indicates that the amplification is physiologically neutral in this infection model (Fig 7E).

However, the results clearly show that amplification does occur in vivo and that copy-number variants are created at frequencies that were never observed during in vitro culturing. Thereby the model indicates that in vivo conditions drive amplification events, leading to genotypic heterogeneity amongst the infecting population.

Additionally, as suggested by this reviewer, we performed animal experiments using a strain carrying a csa1 locus in which every single gene was inactivated (csa1(FS)). Although the locus is repetitive and we show that it is able to amplify under non-selective conditions in vitro (Fig. S3), we observed only a single strain carrying an amplified array after in vivo selection (Fig.7, Ln282-292). This is in strong contrast to WT infections in which we observed amplifications in all but one mice. This strongly suggests that so far unclear pressures selected for heterogeneity of the WT array in vivo.

Minor Points:

1. Fig 1C. The PCR fragment for USFL isolate 165 is not convincing. As this isolate happens to be the one with the highest copy number according to qPCR, an experimental validation of these results would be helpful.

Tandem arrays of csa1 were confirmed using long-read MinION sequencing (Ln131-140, Fig. 1D).

2. The lines in Fig. 2C are not labelled, thus the strength of selection in the evolutionary experiments cannot be assessed.

We apologize for this mistake. Growth curves are now labelled (now Fig. 2B)

Reviewer #2 (Remarks to the Author):

Belikova et al. investigated the gene copy number variations in natural populations of bacteria which causes genotypic and phenotypic heterogeneity in clonal populations of *S. aureus* USA300. The study shows the expansion and also the contraction of gene arrays occur readily in vitro and in vivo with a high various alteration up to more than 100 gene copies in vitro and 30 copies in vivo. This is an interesting topic which explains the tremendous ability of prokaryotes to adapt to changing environments. In most cases selection of point mutations contribute to adaptation; for example respiration defective mutants that lead to small colony variants, SCVs have an advantage by causing persistent infections. Here it was shown that also gene duplication or even multiplication occurred. With the investigation of some representative host colonization and virulence factors such as surface anchored molecule SdrD, the Spl serine proteases and lipoprotein gene sca1, the authors pinpoint the gene amplification occurrence during colonization and infection which is controlled by the balance of the bacterial benefits and the disadvantage to the host.

However, the authors should consider the following points:

Major points:

1. As far as can be seen there was only one pathogen investigated, *S. aureus*, and even not various *S. aureus* strains, but only the *S. aureus* USA300 lineage. This should be then also stated in the title.

This reviewer is correct. We accordingly changed the title to “Gene accordions cause genotypic and phenotypic heterogeneity in clonal populations of Staphylococcus aureus” (Ln1)

2. Lpl3C (SAUSA300_2424) and Lpl2 (SAUSA300_0205) are not lipoprotein genes as they don't contain the right signal peptide with lipobox which is necessary for Igt and Isp function (Shahmirzadi 2016 <https://www.frontiersin.org/articles/10.3389/fmicb.2016.01404/full>). The DOLOP is not the up

to date program to search for the bacterial lipoproteins (Line 110-111). The author should correct them.

We thank the reviewer for pointing this out! This is now correctly stated and the suggested reference is included (In 114-121).

3. *csa1A-D*, *lpl1-9* and tandem *lpp* (SAUZSA300_2429, SAUSA300_2430) share the highly conserved core region 38 aa (Shahmirzadi 2016). However, the author mentioned in Line 394-396 that primers against *csa1* were directed against a highly conserved stretch of the coding sequence to allow amplification of a fragment from all four genes (*csa1A/B/C/D*). The authors should make it clear how they can detect the specific genes for *csa* cluster but other lipoproteins, such as *lpl*.

The DNA sequence of the various genes shows sufficient divergence to allow amplification of csa1 genes but not of the others. The alignment of all genes in the primer binding site is shown in Fig.R2 A in this letter. The primer binding sites are highlighted. qPCR allowed DNA amplification from WT chromosomal DNA while only a diffuse signal was obtained when Δcsa1 DNA was used (Fig.R2 B). We mention this control in the methods section (In486-488.)

Fig. R2. qPCR design for amplification of each gene within the *csa1A-D* locus.

A) Coding sequences of all tandem-lipoproteins were aligned using ClustalW. Identical nucleotides are highlighted in green. Binding sites of qPCR primers used for all experiments are shown in red boxes.

B) Amplification plots of a representative qPCR experiment:

The *ori* control fragment (red) and the *csa1* fragment (blue) were amplified with the respective primer pairs using WT (left) or Δ *csa1* (right) chromosomal DNA.

4. USA300 has 3 different tandem lipoprotein clusters: csa cluster (Csa1A/B/C/D), lpl in vsad island, the rest SAUSA300_2429-2430 should be named differently (such as tandem lpp cluster) to distinguish them.

We changed the designation of the proteins according to this recommendation (ln114-121, FigS2)

Minor points:

1. Line 202-203: instead of REF 20, the author should rather mention the Ref 21 and REF (Stoll 2005) <https://www.ncbi.nlm.nih.gov/pubmed/15784587>.

This was changed accordingly (ln 243-244 Ref 23 + 24)

2. Line 230: ..high csa1 high copy....

This sentence was deleted during restructuring of this paragraph.

3. Line 299: instead of REF 21, the author would rather mention the Ref (Nguyen et al. 2017, <https://www.nature.com/articles/s41467-017-02234-4>).

This was changed accordingly (ln 370-371, Ref 55)

4. Line 406: there is a typo

This was corrected (ln 498)

5. Fig1A III part: there should be 4 genes (csa1A to csa1D) but 5 genes are shown.

This was changed accordingly (Fig1A)

6. Fig 2C lacking the explanation of the CIP concentration for each the growth curve.

We apologize for this mistake. Growth curves are now labelled (now Fig.2B)

7. Fig 2E: lacking the title for Y-axis.

We apologize for this mistake. Y-axis is now labelled (now Fig.4A)

Reviewer #3 (Remarks to the Author):

The title refers to gene copy number variation which is usually a term for multiple numbers of copies of the same gene in a cell, for example those on plasmids or parts of the genome that have replicated to multiple copies. In the text, an alternative is also discussed – gene duplication and amplification (GDA), where a second copy of a gene is introduced into the chromosome downstream of the original copy. These are two different concepts and the terminology used in the title and throughout the manuscript was therefore confusing and conflicting to this reviewer. Are we discussing copies or duplicates? What does the term amplification refer to here?

Our manuscript investigates copy number variation in tandem arrays of genes caused by the “accordion-mechanism” underlying the development of tandem repeats. We are not referring to copy

number variation caused by plasmids or discrete copies of identical genes on different locations on the chromosomes.

We understand that this can cause confusion. Therefore, we have changed the Title to “Gene accordions cause genotypic and phenotypic heterogeneity in clonal populations of Staphylococcus aureus (In 1).

We have also clarified our use of the term “gene copy number variation” (In 48). In addition, we have clarified in several sections in the “abstract” (In 14, 15, 17), “introduction” (In42, 49, 56, 63) and the “results” (In 87, 130, 164) that our experiments aim at detecting tandem-array copy number variation.

The first results section identifying gene copy number in staphylococcal chromosomes is not novel. The concept here described as GDA is well known in *S. aureus* chromosomes for a range of genes, particularly in the pathogenicity islands (examples Jarraud et al., J Immunol 2001; Al Shangiti et al., Infect Immun 2004). Although these genes were likely generated by duplication, each individual gene has a unique function, eg. Ssls, This would argue that many GDAs are stable, and functional. Another similar and well described duplication is in the repetitive binding units within genes, such as *spaA* used for *S. aureus* typing where one or two duplications arising in a short period are seen (Boye & Westh, FEMS Micro Lett 2011). In all cases, the maximum number of duplicates is around 20 and does not normally arise from bacterial subculture. This context would be useful for the paper. The authors should also consider discussion of the limitation of using short read sequencing to estimate gene copy number.

We agree that repetitive arrays within S. aureus are known and that S. aureus isolates from different lineages contain different number of repeats. The novelty of our analysis and findings is in observing heterogeneity in a set of USA300 strains of close temporal and spatial origin. Thereby our results challenge the hypothesis that arrays of repetitive, yet functional diverse genes are stable. We suggest that such arrays are still expanding/contracting to create further diversity and heterogeneity even in clonal populations.

*We agree, spa appears stable and is successfully used to distinguish lineages. Yet, the repetitive signatures in spa consist of ~8-11 repeats of a 24bp signature [3], giving a total length of ~192-264 bp. RecA is responsible for many GDAs but needs ~100bp of homology between two DNA fragments to allow recombination. Therefore, it is questionable whether creation of duplication/deletion within the spa-repeats is due to the same amplification/deletion mechanisms that generate variation in the loci investigated in this manuscript (4~700 bp-long repeated *csa1* genes). Even if RecA was responsible, the frequency of spa-alteration would be significantly lower than those observed for regions possessing longer repeats.*

However, using our datasets, we cannot make any further conclusion about the stability of spa as the set of isolates used for the original whole genome sequencing was chosen according to its spa-type [4]. Therefore, isolates are identical and no conclusions regarding the stability of spa can be drawn.

As suggested by this reviewer, we additionally performed MinION sequencing (see below) and now discuss these results in light of the ones obtained by short-read sequencing (In 321-326).

The novel Results begin by focusing on a single gene, *csa1* and showing that GDA duplicates can be seen in different isolates. Figure 1 justifies the use of qPCR as a method for identifying the numbers of duplicate *csa1* genes in isolates with 1-4 duplicates.

Figure 2a. shows evidence of *csa1* duplication after standard growth in the laboratory. qPCR amplification suggests over 100 duplications. But I do not see the evidence that *csa1* gene has duplicated 100 times. SMRT or minlon sequencing showing the duplications would be useful. Without this evidence, it suggests that qPCR is not be a robust method for detecting large numbers of duplications.

We performed MinION sequencing (methods In 450-459) on several independent high copy number variants of csa1 (Fig 3B) and sdrD (new Fig 4D) as well as on the USFL isolates of the initial screen (Fig 1D). The long-read sequencing data confirm the qPCR estimates and unequivocally demonstrate that tandem amplification of the csa1 and sdrD genes occurred. In high copy number isolates such as C6 (~150 copies of csa1 measured by qPCR) and L38 (~80 copies measured by qPCR), even MinION reads longer than 110kb failed to stretch the entire tandem array of csa1, suggesting a csa1 copy number of at least 70 (Fig 3B). In these cases, we dare saying that qPCR appears even more powerful than MinION sequencing do discriminate these copy number variants. The sequencing is described in In 164-176 and In 216-219.

Since the authors propose multiple genes are duplicated, and this occurs in normal growth conditions, why are whole genome sequencing using SMRT technology not identifying isolates with 100 or more duplications of multiple genes? Could the qPCR method be amplifying mRNA generating artificially high counts in the experiments? Or that some regions of the chromosomal DNA are replicating at higher frequency?

Our Minlon-sequencing data confirm tandem amplification of the arrays and indicate that our qPCR analysis is robust.

We assume that in current “long read sequencing” projects tandem amplifications are not appreciated, as they are only present in a minor fraction of cells. Our experiments suggest that copy number variation within e.g. the csa1 locus occurs in 1 in 10⁸ cells (under nonselective in vitro conditions). Even if their presence was by magnitudes higher under selective conditions in vivo, tandem amplification would very likely be missed when only individual clones from patients or environmental samples are sequenced. We believe that, in the future, only batch sequencing approaches, whereby thousands of clones recovered from a single patient/sample are sequenced together in a very deep fashion might truly help to trace copy number variation in tandem arrays. Of note, a recent article deposited on bioarchives detected genetic accordions in individual cells of a clonal population of Bordetella pertussis [5]. Similarly, we provide evidence that individual cells in clonal populations of csa1 (In 171-176 Fig. 3) and sdrD (In 216-219, Fig 4D) high copy number variants differed in gene number.

Additionally, the effects of selective pressures acting on tandem amplifications must not be overlooked. In our in vitro experiments we recovered variants with ~100 copies while in vivo the highest copy number recovered was ~30. Moreover, strains harboring ~100 copies were not more virulent than strains harboring 4 copies in our models. Hence, it remains unclear whether selective pressures in vivo counteract the development of a certain length of the arrays and additional experiments are needed to address these questions.

In Figure 2b the y-axis shows very low numbers but it is not clear from the legend or the methods how this experiment was performed. What is being measured?

Details are given in the methods section (In 501-504). We also added information to the Figure legend (Fig 2C, 847-848).

Briefly: We grew the csa1::tetK strain in the presence or absence of ciprofloxacin for three consecutive cultures. Each day a sample was taken and plated on TSA (total living counts) as well as on TSA_{Tc20} (putative amplifications). Subsequently, up to 20 tet-resistant colonies were screened for csa1 copy number.

The data shown in 2C reflect the frequency of amplification (Tc20 resistant clones that showed at least a 2-fold increase in csa1 copy number by qPCR) within the total population of living cells in 6 independent cultures.

From here on there is such a doubt raised about the validity of the qPCR to detect gene duplication

that it is not clear how the rest of the data should be interpreted.

Therefore, while the concept of this paper is interesting, the results are preliminary without the necessary robust evidence that the genes described are in fact duplicating.

References:

1. Nguyen MT, Kraft B, Yu W, Demircioglu DD, Hertlein T, Burian M, Schmalzer M, Boller K, Bekeredjian-Ding I, Ohlsen K *et al*: **The nuSaalpha Specific Lipoprotein Like Cluster (lpl) of S. aureus USA300 Contributes to Immune Stimulation and Invasion in Human Cells.** *PLoS Pathog* 2015, **11**(6):e1004984.
2. Shang W, Rao Y, Zheng Y, Yang Y, Hu Q, Hu Z, Yuan J, Peng H, Xiong K, Tan L *et al*: **beta-Lactam Antibiotics Enhance the Pathogenicity of Methicillin-Resistant Staphylococcus aureus via SarA-Controlled Lipoprotein-Like Cluster Expression.** *MBio* 2019, **10**(3).
3. Koreen L, Ramaswamy SV, Graviss EA, Naidich S, Musser JM, Kreiswirth BN: **spa typing method for discriminating among Staphylococcus aureus isolates: implications for use of a single marker to detect genetic micro- and macrovariation.** *J Clin Microbiol* 2004, **42**(2):792-799.
4. Uhlemann AC, Dordel J, Knox JR, Raven KE, Parkhill J, Holden MT, Peacock SJ, Lowy FD: **Molecular tracing of the emergence, diversification, and transmission of S. aureus sequence type 8 in a New York community.** *Proc Natl Acad Sci U S A* 2014, **111**(18):6738-6743.
5. Abrahams JS, Weigand MR, Ring N, MacArthur I, Peng S, Williams MM, Bready B, Catalano AP, Davis JR, Kaiser MD *et al*: **Duplications drive diversity in *Bordetella pertussis* on an underestimated scale.** *bioRxiv* 2020:2020.2002.2006.937284.

REVIEWERS' COMMENTS:

Reviewer #4 (Remarks to the Author):

It was a pleasure reading the revised manuscript by Belikova et al. As I was asked to review a revision of the paper, I focused on whether the authors responded well to reviewers' comments, whether there are any major problems that had been overlooked before, and also whether there are some minor errors left. Overall, I find the study of high general importance. It provides convincing evidence for the importance of scalable gene amplification or gene accordeons during host adaptation of *Staph. aureus* and the likely underlying reasons. The study was very well set-up, starting with a broad description of the phenomenon in clinical isolates, followed by a detailed experimental analysis using evolution experiments, functional genetic analysis, and also infection experiments in mice. The specific results are based on well designed experiments and a sound statistical analysis of the data. The conclusions drawn are very well justified by the data. The functional importance of such gene amplifications is usually unknown and the phenomenon itself often neglected in current characterizations of human pathogens. Thus, the study by Belikova et al. fills an important knowledge gap and consequently it is highly suited for publication in *Nature Communications*.

I do not see any major problems in the revised manuscript. Moreover, in my opinion, the authors did address the previously raised concerns very well. I only have two minor comments:

- 1) Figure 2: The authors should briefly add an explanation on why ciprofloxacin was used in the experiments. This is not clear. Moreover, Fig. 2B needs further explanation. It is not clear what the curves should highlight and what exactly is shown in the two rows of the Western blot.
- 2) Lines 248 onwards: To enhance clarity, the authors should in this part of the manuscript explain the rationale of their experiment in more detail. In particular, they should explain why the supernatant of the bacterial cultures is used to assess the immunestimulatory effect of the *csa1*-locus encoded lipoproteins. Is it known that these lipoproteins are shedded and thus available in the supernatant?

Reviewer #5 (Remarks to the Author):

The study by Balikova et al. analyses tandem amplifications or deletions of genes in natural bacterial populations and bacterial lipoproteins potential to modulate host pathogen interactions through amplifications/segregations.

The authors show that tandem amplifications of the gene *csa1* in *S. aureus* genome sequences are common. However, for other genes deletions are dominating. They show that amplifications and segregations of the tandem amplifications of *csa1* is happening at a high frequency both in vivo and in vitro. With the additional negative controls (inactivated *csa1* gene, (FS)) the experiments are convincingly showing that the altered copy number of *csa1* changes the gene expression and as such modulates the immune response.

The manuscript is well written and has been improved with additional crucial experiments as suggested by previous reviewers (inactivation of *csa1*). This paper will add to the field of unstable amplifications that transiently changes gene expression.

Minor points.

1. The authors state that they found several hotspots of variation, gene copy number (stated in the abstract). According to the supplemental information most of the hotspots contained deletions of genes and very few amplifications with the exception of *csa1*. As deletions are evolutionary dead ends when it comes to amplification of tandem repeats, which also the authors recognize, the impact of multiple hotspots changing gene expression is over stretching the message as written in

the abstract.

2. The sentence "Interestingly, we observed several spontaneous segregation events where by cells of the culture had undergone deletions that led to a drastic reduction in copy number (eg E26A and E26B in Fig. 5A)" make no sense to me.

I. I can't find information on E26A and E26B in Fig. 5A, do they mean Fig. 6A.

II. I don't see reduction of E26B in Fig 6A (if that is the correct figure), if anything an increase is detected. Maybe they mean L38A?

Response to referees' comments

Reviewer's Comments:

Reviewer #4 (Remarks to the Author)

It was a pleasure reading the revised manuscript by Belikova et al. As I was asked to review a revision of the paper, I focused on whether the authors responded well to reviewers' comments, whether there are any major problems that had been overlooked before, and also whether there are some minor errors left. Overall, I find the study of high general importance. It provides convincing evidence for the importance of scalable gene amplification or gene accordions during host adaptation of *Staph. aureus* and the likely underlying reasons. The study was very well set-up, starting with a broad description of the phenomenon in clinical isolates, followed by a detailed experimental analysis using evolution experiments, functional genetic analysis, and also infection experiments in mice. The specific results are based on well designed experiments and a sound statistical analysis of the data. The conclusions drawn are very well justified by the data. The functional importance of such gene amplifications is usually unknown and the phenomenon itself often neglected in current characterizations of human pathogens. Thus, the study by Belikova et al. fills an important knowledge gap and consequently it is highly suited for publication in Nature Communications.

I do not see any major problems in the revised manuscript. Moreover, in my opinion, the authors did address the previously raised concerns very well. I only have two minor comments:

We thank this referee for his/her positive comments.

1) Figure 2: The authors should briefly add an explanation on why ciprofloxacin was used in the experiments. This is not clear. Moreover, Fig. 2B needs further explanation. It is not clear what the curves should highlight and what exactly is shown in the two rows of the Western blot.

The rationale behind the use of ciprofloxacin was mentioned in the text (lines 202-206). This is now explained also in the legend of Figure 2B. As ciprofloxacin inhibits bacterial growth and induces RecA expression, we quantified both growth rate (curves) and RecA protein levels (Western Blot). We now clarify this in the legend of Figure 2B.

2) Lines 248 onwards: To enhance clarity, the authors should in this part of the manuscript explain the rationale of their experiment in more detail. In particular, they should explain why the supernatant of the bacterial cultures is used to assess the immunestimulatory effect of the *csa1*-locus encoded lipoproteins. Is it known that these lipoproteins are shedded and thus available in the supernatant?

Yes, we thank the reviewer for this comment! We modified the text to clarify that Csa1 molecules are shed from the cell surface in a PSM-dependent manner and become available in the cell-free supernatant which is the basis for immunostimulatory activity [1]. We also added this reference to the text (lines 269-276).

Reviewer #5 (Remarks to the Author)

The study by Balikova et al. analyses tandem amplifications or deletions of genes in natural bacterial populations and bacterial lipoproteins potential to modulate host pathogen interactions through amplifications/segregations.

The authors show that tandem amplifications of the gene *csa1* in *S. aureus* genome sequences are common. However, for other genes deletions are dominating. They show that amplifications and segregations of the tandem amplifications of *csa1* is happening at a high frequency both in vivo and in vitro. With the additional negative controls (inactivated *csa1* gene, (FS)) the experiments are convincingly showing that the altered copy number of *csa1* changes the gene expression and as such modulates the immune response.

The manuscript is well written and has been improved with additional crucial experiments as suggested by previous reviewers (inactivation of *csa1*). This paper will add to the field of unstable amplifications that transiently changes gene expression.

We thank this referee for his/her positive comments.

Minor points.

1. The authors state that they found several hotspots of variation, gene copy number (stated in the abstract). According to the supplemental information most of the hotspots contained deletions of genes and very few amplifications with the exception of *csa1*. As deletions are evolutionary dead ends when it comes to amplification of tandem repeats, which also the authors recognize, the impact of multiple hotspots changing gene expression is over stretching the message as written in the abstract.

*In lines 376-383 we discuss why our sequence analysis detected more frequently loci with gene deletion rather than amplification: Firstly, amplifications, due to their intrinsic instability, could be lost in the clinical isolates following isolation and in vitro cultivation. Secondly, we demonstrate that loci only associated with deletions in clinical isolates (*SdrD*) undergo amplification in vitro supporting the hypothesis that amplification and deletion are always co-occurring phenomena. Amplification of e.g. *SdrD* might be beneficial only in very specific environmental niches (discussed in the text lines 425-442) while it might represent a burden under the “conventional” *S. aureus* isolation-sites sampled by Uhlemann and colleagues. This might explain why we did only identify deletions and not amplifications.*

As such we believe that referring to several hotspots of copy number variation is justified.

2. The sentence “Interestingly, we observed several spontaneous segregation events where by cells of the culture had undergone deletions that led to a drastic reduction in copy number (eg E26A and E26B in Fig. 5A)” make no sense to me.

I. I can't find information on E26A and E26B in Fig. 5A, do they mean Fig. 6A.

II. I don't see reduction of E26B in Fig 6A (if that is the correct figure), if anything an increase is detected. Maybe they mean L38A?

*We thank the reviewer for pointing out the mistake: We indeed referred to figure 6A (now changed in line 281). As indicated by the filled dark dots in this figure, the *csa1* copy number of two independent cultures of the same lineage, e.g. E26A and E26B, greatly differed on the day of the experiment, with E26A displaying low copy number and E26B displaying high copy number as did the original lineage*

(shown in Figure 3B). We have now better clarified this in the text (lines 278-282) as well as in the legend of Figure 6A.

1. Hanzelmann, D., et al., *Toll-like receptor 2 activation depends on lipopeptide shedding by bacterial surfactants*. Nat Commun, 2016. **7**: p. 12304.